# New Paradigm of Adversarial Training: Releasing Inherent Trade-Off between Accuracy and Robustness via Dummy Classes

## Abstract

Adversarial Training (AT) is recognized as one of the most effective methods to enhance the robustness of Deep Neural Networks (DNNs). However, existing AT methods suffer from an inherent trade-off between adversarial robustness and clean accuracy, which seriously hinders their real-world deployment. Previous works have studied this trade-off within the current AT paradigm, exploring various factors such as perturbation intensity, label noise and class margin. Despite these efforts, current AT methods still typically experience a reduction in clean accuracy by over 10% to date, without significant improvements in robustness compared with simple baselines like PGD-AT. This inherent trade-off raises a question: whether the current AT paradigm, which assumes to learn the corresponding benign and adversarial samples as the same class, inappropriately combines clean and robust objectives that may be essentially inconsistent. In this work, we surprisingly reveal that up to 40% of CIFAR-10 adversarial samples always fail to satisfy such an assumption across various AT methods and robust models, explicitly indicating the improvement room for the current AT paradigm. Accordingly, to relax the tension between clean and robust learning derived from this overstrict assumption, we propose a new AT paradigm by introducing an additional dummy class for each original class, aiming to accommodate the hard adversarial samples with shifted distribution after perturbation. The robustness *w.r.t.* these adversarial samples can be achieved by runtime recovery from the predicted dummy classes to their corresponding original ones, eliminating the compromise with clean learning. Building on this new paradigm, we propose a novel plug-and-play AT technology named **DU**mmy **C**lasses-based **A**dversarial **T**raining (**DUCAT**). Extensive experiments on CIFAR-10, CIFAR-100, and Tiny-ImageNet demonstrate that the DUCAT concurrently improves clean accuracy and adversarial robustness compared with state-of-the-art benchmarks, effectively releasing the existing inherent trade-off. The code is available at *https://anonymous.4open.science/r/DUCAT*.

## 1 Introduction

Deep Neural Networks (DNNs) have demonstrated remarkable performance in various real-world applications, but they remain vulnerable to adversarial attacks (Biggio et al., 2013; Szegedy et al., 2014). Specifically, malicious input perturbations that are often imperceptible to humans can cause significant changes in the output of DNNs (Goodfellow et al., 2015; Huang et al., 2020), raising serious security concerns within both the public and research communities (Wang et al., 2020b). In response, several defense mechanisms have been proposed to enhance the adversarial robustness of DNNs, such as *defense distillation* (Papernot et al., 2017), *feature squeezing* (Xu et al., 2018), *randomization* (Xie et al., 2018), and *input denoising* (Guo et al., 2018; Liao et al., 2018). However, most of these techniques have proven subsequently to rely on *obfuscated gradients* (Athalye et al., 2018) and be ineffective against more advanced adaptive attacks (Tramer et al., 2020).

Currently, Adversarial Training (AT) (Goodfellow et al., 2015; Madry et al., 2018) is demonstrated as one of the most effective approaches to train inherently robust DNNs (Athalye et al., 2018; Dong et al., 2020). Different from clean training, where Empirical Risk Minimization (ERM) serves as the fundamental paradigm, AT directly utilizes adversarially augmented samples to yield robust models.

Figure 1: Comparison between the proposed DUCAT under our new AT paradigm and PGD-AT under the current one. Previously, AT assumes to learn each crafted adversarial $\mathbf{x}'$ with the same $\mathbf{y}$ as the corresponding benign $\mathbf{x}$, aiming at directly classifying unseen $\mathbf{x}'$ from potential inference-time adversaries to the correct class. In contrast, we suggest $C$ more dummy classes, along with a uniquely designed two-hot soft label-based learning to one-to-one bridge these dummy classes with original ones. In this way, some hard $\mathbf{x}'$ with shifted distribution can be accommodated without significantly hurt clean learning on $\mathbf{x}$, and their robustness can be ensured by recovery from predicted $[C+1 \ldots 2C]$ to original $[1 \ldots C]$. Relaxing the current overstrict assumption, our new AT paradigm releases the inherent trade-off between accuracy and robustness.

Research indicates that this paradigm is equivalent to optimizing an upper bound of natural risk on the original data, and can thereby serve as a principle against adversarial attacks (Tao et al., 2021). To the best of our knowledge, despite considerable advancements in the specific mechanism of existing AT methods, most of them remain adhering to this principal paradigm. The solid AT benchmarks such as PGD-AT (Madry et al., 2018), TRADES (Zhang et al., 2019) and MART (Wang et al., 2020b) are representative examples.

However, recent studies indicate that existing AT methods suffer from an inherent trade-off between adversarial robustness and clean accuracy (Tsipras et al., 2019; Zhang et al., 2019; Raghunathan et al., 2019; 2020; Wang et al., 2020a; Bai et al., 2021). This means improving robustness via these AT methods is at the cost of reducing model accuracy compared with the standard training, which can seriously hurt the experience of benign users and greatly reduce the use of AT among real-world DNN application providers. A widely recognized cause of this problem is that AT equally requires predictive consistency within the $\epsilon$-ball of each sample, which can complicate decision boundaries, particularly for those samples near class margins, ultimately degrading clean generalization (Dong et al., 2022; Rade & Moosavi-Dezfooli, 2022; Cheng et al., 2022; Yang & Xu, 2022). To learn robustness *w.r.t.* such samples more appropriately, FAT (Zhang et al., 2020) and HAT (Rade & Moosavi-Dezfooli, 2022) utilized adaptive perturbations in AT to smooth decision boundaries; Consistency-AT (Tack et al., 2022) encouraged the similarity of predictive distributions between adversarial samples derived from different augmentations of the same instance, thereby enhancing learnable patterns and reducing label noise; SOVR (Kanai et al., 2023) explicitly increased *logits* margin of important samples by switching from *cross-entropy* to a new one-vs-the-rest loss.

Despite several previous efforts, there has been limited substantial progress in addressing this trade-off problem. Most impressive advancements in recent years come from those methods introducing extra data (Alayrac et al., 2019; Carmon et al., 2019; Najafi et al., 2019; Zhai et al., 2019; Li et al., 2022) or utilizing generative models (Wang et al., 2023b). However, they not only demand more resources and computational costs but also violate the conventional fairness assumption of AT (*i.e.*, no additional data should be incorporated (Pang et al., 2021)). Other advanced AT techniques apart from these still typically experience a drop in clean accuracy exceeding 10% to date, and the state-of-the-art (SOTA) robustness has just improved slowly (*i.e.*, about 1% per year on average since 2018) (Wei et al., 2023; Dong et al., 2023; Li & Spratling, 2023; Jia et al., 2024). This inherent trade-off raises a question about the current AT paradigm on uniformly learning accuracy and robustness:

*Is the current AT paradigm, which compels DNNs to classify corresponding benign and adversarial samples into the same class, really appropriate and necessary for achieving adversarial robustness?*

In this work, as straightforward evidence supporting our deduction, we reveal that certain samples that **always fail** to meet the above objective of the current AT paradigm generally exist across various AT methods and different robust models. This observation implies that the conventional assumption

of categorizing adversarial samples to the same class as the corresponding benign ones may be over-strict. As such, with the learning of benign and adversarial samples as two essentially inconsistent targets, the attempt to unify them in the current AT paradigm can be improper, which is likely to be blamed for the existing trade-off between clean accuracy and adversarial robustness. In response, we propose a new AT paradigm introducing additional dummy classes for certain adversarial samples that differ in distribution from the original ones to relax the current overstrict assumption. Then accordingly, we propose a novel AT method called **DU**mmy **C**lasses-based **A**dversarial **T**raining (**DUCAT**), releasing the inherent trade-off between clean accuracy and adversarial robustness.

Our core idea is to create dummy classes with the same number as the original ones and respectively attribute them as the primary targets for benign and adversarial samples during training. Importantly, we do not suggest a strict separation between corresponding benign and adversarial samples (*i.e.*, to make the original classes completely benign and the dummy ones completely adversarial), because assuming corresponding benign and adversarial samples have completely different distributions is still excessive, which may cause memorization of hard-label training samples, resulting in overfitting to specific adversaries as demonstrated by our toy case in Section 2.2.3. Instead, we construct unique two-hot soft labels to explicitly bridge corresponding original and dummy classes as the suboptimal alternative target of each other, so that the separation also becomes learnable, to utilize the potential of DNNs further. During inference time, such one-to-one correspondences enable the detection and recovery of adversarial samples, thereby achieving robustness without significantly compromising clean learning. Specifically, if a test sample is classified into a dummy class, this serves as an indication of a potential adversarial attack, allowing us to recover its clean prediction through a projection back to the corresponding original class. Separated from the computation graph of DNN and implemented in a run-time-only manner, such a projection further degrades the ability of real-world adversaries.

**Contributions. 1)** For the first time, we empirically reveal that **always-failed** samples widely exist in conventional AT, explicitly suggesting the assumption of the current AT paradigm to learn benign and adversarial samples with the same labels is **overstrict**; **2)** We propose a **new AT paradigm** introducing dummy classes to relax the current assumption, **releasing** the inherent trade-off between accuracy and robustness from it; **3)** A novel **plug-and-play DUCAT** method is proposed, concurrently improving the accuracy and robustness of four common AT benchmarks in large-scale experiments on CIFAR-10, CIFAR-100 and Tiny-ImageNet, significantly outperforming 16 SOTAs.

## 2 INVOLVING DUMMY CLASSES IN ADVERSARIAL TRAINING

### 2.1 MOTIVATION: **ALWAYS-FAILED** SAMPLES WIDELY EXIST IN AT

The undesirable progress regarding the trade-off between clean accuracy and adversarial robustness in the past years makes us suspect that the assumption of the current AT paradigm to assign corresponding benign and adversarial samples to the same class may be inappropriate and un-necessary. As intuitive evidence of this deduction, by two proof-of-concept experiments, we demonstrate that up to 40% adversarial samples that **always fail** to meet such an assumption generally exist in conventional AT crossing various AT methods and different robust models. The experiments are conducted on CIFAR-10 (Krizhevsky & Hinton, 2009) and ResNet-18 (He et al., 2016), with, as mentioned in Section 1, the most popular AT benchmarks, PGD-AT, TRADES, MART, and a representative SOTA, Consitency-AT. The training details are the same as our main experiments in Section 3.1. For adversary, we adopt a typical and effective adversarial attack, PGD-10 (Madry et al., 2018).

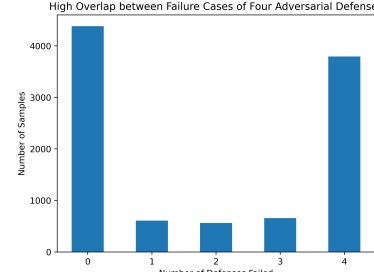

Figure 2: High overlap between adversarial samples evading from four AT benchmarks, implying such failures are more likely due to an inappropriate learning objective from overstrict assumption of the current AT paradigm, rather than any specific AT methods.

**Always-failed cases crossing AT methods.** We first demonstrate that there is a high overlap in both successful and unsuccessful cases of the four AT methods. Specifi-

cally, for every benign test sample, with a high probability (*i.e.*, > 80%), its corresponding adversarial sample can either be uniformly defended by the four methods or escape from all of them. As illustrated in Figure 2, we report the numbers of samples that can respectively beat none, one, two, three and all experimental defenses. The samples on which all the defenses are uniformly worked or failed make up the overwhelming majority. Besides, the confusion matrixes of the models protected by these four AT methods under the adversary are shown in Figure 8 in Appendix C, where similar failure patterns can be noted crossing all of them. Based on these observations, we state that *always-failed adversarial samples widely exist no matter the specific choice of AT methods*.

**Always-failed cases crossing robust models.** By respectively adopting the four AT benchmarks to train a model from scratch and then test it by the same white-box adversary, above we have revealed that the always-failed cases are independent of specific AT methods. In this part, through black-box transfer attacks, we further demonstrate that such cases should not be attributed to the weakness of any specific robust models as well. To be specific, we first train four different robust models respectively by the benchmarks and select any one of them as the surrogate model, based on which we generate adversarial samples for the transfer attack. Then we divide the generated samples into two subsets based on whether successfully attack the surrogate model, and respectively use them to attack the other three models. Figure 3 shows the results with the four robust models serving as the surrogate one in order, where significant differences in transferability between the originally successful

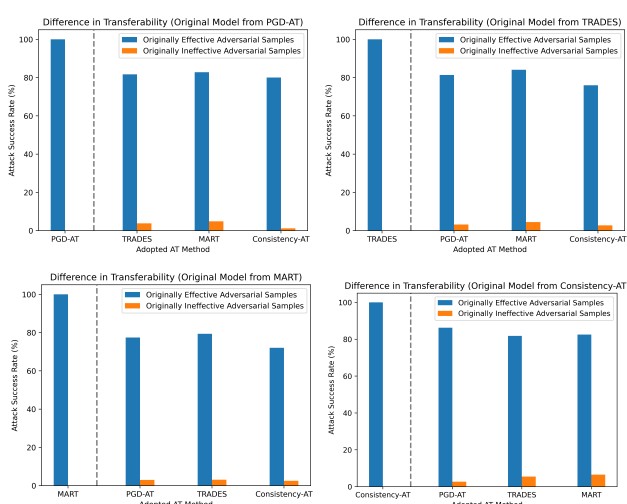

Figure 3: The robust models already enhanced by different AT methods are still highly likely to be uniformly beaten by the successful adversarial samples generated based on any one of these models. Such a deconstruction of adversarial transferability between robust models reveals the model-independent vulnerability of certain samples, which further supports our deduction for the current AT paradigm.

and unsuccessful subsets can be clearly observed. In other words, for every single adversarial sample generated based on any robust model, if it is originally effective/ineffective in attacking this surrogate model, then with a high probability, it would also work/fail on other robust models. This implies that *always-failed cases generally exist regardless of the specific robust model as the target*.

Now that the wide existence of always-failed samples is confirmed crossing different AT methods and robust models, compared with simply attributing this to specific technology or implementation details, it seems more reasonable to rethink if it is the paradigm to be blamed in essence. Based on the analysis in this section, it is not difficult to notice that *always learning to classify adversarial samples to the same class as the corresponding benign ones is likely to be a too-hard assumption*. In that case, trying to compulsively unify the learning of benign and adversarial samples, which may be two essentially inconsistent optimization objectives, becomes a logical and reasonable source of the existing trade-off between clean accuracy and adversarial robustness. This inspired us to propose a new AT paradigm in which the overstrict current assumption can be relaxed.

## 2.2 NEW ADVERSARIAL TRAINING PARADIGM WITH DUMMY CLASSES

A natural idea to relax the current assumption is to introduce additional dummy classes to accommodate the adversarial samples hard to classify to the original class, based on which we propose our new AT paradigm. In this section, we first provide preliminaries *w.r.t.* relevant concepts, followed by a formal formulation of our paradigm. Then via a toy case deliberately causing overfitting to the training adversary, we demonstrate the necessity of designing two-hot soft labels than hard ones.

### 2.2.1 PRELIMINARIES

In the context of clean training, for a classification task with $C \geq 2$ as the number of classes, given a dataset $\mathcal{D} = \{(\mathbf{x}_i, y_i)\}_{i=1,...,n}$ with $\mathbf{x}_i \in \mathbb{R}^d$ and $y_i \in \{1, ..., C\}$ respectively denoting a natural sample and its supervised label, as well as $\mathbf{y}_i \in \mathbb{N}^C$ being the *one-hot* format of $y_i$, ERM aims to learn a classifier $h_{\boldsymbol{\theta}} : \mathbf{x} \to \mathbb{R}^C$, such that $h_{\boldsymbol{\theta}}(\mathbf{x}_i) = \mathbf{q}(\mathbf{x}_i, \boldsymbol{\theta})$ represents the output *logits* with each element $\mathbf{q}^{(k)}(\mathbf{x}_i, \boldsymbol{\theta})$ corresponding to class $k$, to minimize the empirical risk $\mathbb{E}_{\mathbf{x}, \mathbf{y} \sim \mathcal{D}} [\mathcal{L}(h_{\boldsymbol{\theta}}(\mathbf{x}), \mathbf{y})]$ through a classification loss $\mathcal{L}$ like *cross-entropy* (*CE*). Such a learning objective is formulated as:

$$\boldsymbol{\theta}^* = \arg\min_{\boldsymbol{\theta}} \frac{1}{n} \sum_{i=1}^{n} \mathcal{L}(h_{\boldsymbol{\theta}}(\mathbf{x}_i), \mathbf{y}_i). \tag{1}$$

In contrast, AT augments conventional ERM to $\mathbb{E}_{\mathbf{x}', \mathbf{y} \sim \mathcal{D}} [\max_{\mathbf{x}' \in \mathcal{P}(\mathbf{x})} \mathcal{L}(h_{\boldsymbol{\theta}}(\mathbf{x}'), \mathbf{y})]$, where $\mathbf{x}'$ denotes the adversarial sample *w.r.t.* $\mathbf{x}$ and $\mathcal{P}(\mathbf{x})$ is a pre-defined perturbation set, to approximate the minimal empirical loss even under the strongest attack, directly learning the concept of robustness (Madry et al., 2018). Thus, the current AT objective can be formulated as a *min-max* problem :

$$\boldsymbol{\theta}^* = \arg\min_{\boldsymbol{\theta}} \frac{1}{n} \sum_{i=1}^{n} \max_{\mathbf{x}'_i \in \mathcal{P}(\mathbf{x}_i)} \mathcal{L}(h_{\boldsymbol{\theta}}(\mathbf{x}'_i), \mathbf{y}_i). \tag{2}$$

On the other hand, to our knowledge, the only previous work that involves the concept of dummy class is DuRM (Wang et al., 2023a), which aims to develop a general, simple and effective improvement to ERM for better generalization in various tasks. Specifically, in response to an assumption that ERM generalizes inadequately when the existence of outliers increases uncertainty and varies training and test landscapes (Cha et al., 2021), DuRM enlarges the dimension of output logits, providing implicit supervision for existing classes and increasing the degree of freedom. Formally, given $(\cdot \| \cdot)$ denoting *concat*, DuRM proposed to add $C_d$ dummy classes and transfer the original $C$-class classification task to a $(C + C_d)$-class classification problem:

$$\boldsymbol{\theta}^* = \arg\min_{\boldsymbol{\theta}} \frac{1}{n} \sum_{i=1}^{n} \mathcal{L} \left( h_{\boldsymbol{\theta}}(\mathbf{x}_i) \| h_{\boldsymbol{\theta}}^{\text{DuRM}}(\mathbf{x}_i), \mathbf{y}_i \| \mathbf{0} \right), \tag{3}$$

where $h_{\boldsymbol{\theta}}^{\text{DuRM}}(\mathbf{x}_i) \in \mathbb{R}^{C_d}$ denotes the output *logits* of DuRM and $\mathbf{0}$ is a zero vector with the same dimension, which means there is no supervision information for these additional dummy classes.

### 2.2.2 PARADIGM FORMULATION

Provided the statement in Section 2.1 that the current assumption in conventional AT paradigm to learn adversarially perturbed samples with the same supervised label as the corresponding benign ones is likely to be a too-hard optimization objective, we define a novel AT paradigm to relax this assumption. Formally, for a $C$-class classification task, we propose to append another $C$ dummy classes to build **one-to-one correspondence** between original classes and the dummy ones. Then to optimize this new $2 \cdot C$-class classification problem, we introduce dummy label $\dot{\mathbf{y}}_i \| \ddot{\mathbf{y}}_i$ such that:

$$\dot{\mathbf{y}}_i^{(k)} + \ddot{\mathbf{y}}_i^{(k)} = \begin{cases} 1, & k = y_i \\ 0, & k \neq y_i \end{cases}, \tag{4}$$

where $\dot{\mathbf{y}}_i + \ddot{\mathbf{y}}_i$ equals to the original *one-hot* label vector $\mathbf{y}_i$. Then with $\mathbf{x}'$ being the adversarial sample generated from $\mathbf{x}$ and $h_{\boldsymbol{\theta}}^{\text{Dummy}}(\mathbf{x}_i) \in \mathbb{R}^C$ denoting the output *logits* from the $C$ appended dummy classes, our new AT paradigm can be formulated as:

$$\boldsymbol{\theta}^* = \arg\min_{\boldsymbol{\theta}} \frac{1}{n} \sum_{i=1}^{n} \left( \mathcal{L} \left( h_{\boldsymbol{\theta}}(\mathbf{x}_i) \| h_{\boldsymbol{\theta}}^{\text{Dummy}}(\mathbf{x}_i), \dot{\mathbf{y}}_i \| \ddot{\mathbf{y}}_i \right) + \max_{\mathbf{x}'_i \in \mathcal{P}(\mathbf{x}_i)} \mathcal{L} \left( h_{\boldsymbol{\theta}}(\mathbf{x}'_i) \| h_{\boldsymbol{\theta}}^{\text{Dummy}}(\mathbf{x}'_i), \dot{\mathbf{y}}'_i \| \ddot{\mathbf{y}}'_i \right) \right), \tag{5}$$

where $\dot{\mathbf{y}}'_i$ and $\ddot{\mathbf{y}}'_i$ can be differently weighted with $\dot{\mathbf{y}}_i$ and $\ddot{\mathbf{y}}_i$. For example, it would be valid to assign that $\dot{\mathbf{y}}_i^{(k)} + \ddot{\mathbf{y}}_i^{(k)} = 0.5 + 0.5 = 1$ while $\dot{\mathbf{y}}_i'^{(k)} + \ddot{\mathbf{y}}_i'^{(k)} = 1 + 0 = 1$.

At inference time, we can acquire the final output of the robust DNN classifier by projecting each dummy class to the corresponding original ones. As such a projection is separated from the computation graph of DNN and can be merely implemented in the run time, the robust mechanism can

be simply explained as detecting (possible) adversarial attacks when certain samples are classified to the dummy classes and recovering such samples back to the corresponding original classes. We defer detailed discussion on the real-world threat model including adversary capacity to Section 3.1. Formally, with $\hat{y}$ denoting the final predicted class label, the projection can be formulated as:

$$\hat{y} = \begin{cases} g_{\boldsymbol{\theta}}(\mathbf{x}_i), & g_{\boldsymbol{\theta}}(\mathbf{x}_i) \leq C \\ g_{\boldsymbol{\theta}}(\mathbf{x}_i) - C, & g_{\boldsymbol{\theta}}(\mathbf{x}_i) > C \end{cases} \quad \text{where} \quad g_{\boldsymbol{\theta}}(\mathbf{x}_i) = \arg\max_{k=1,\ldots,2 \cdot C} \left( h_{\boldsymbol{\theta}}(\mathbf{x}_i) \,\|\, h_{\boldsymbol{\theta}}^{\text{Dummy}}(\mathbf{x}_i) \right)^{(k)}. \quad (6)$$

Our new paradigm explicitly distinguishes the learning of benign and adversarial samples, relaxing the overstrict assumption in the current AT paradigm that always pursues consistent class distribution for them. This is expected to release the unnecessary tension between the standard and robust optimization objectives, and as a result, release the trade-off between clean accuracy and robustness observed in existing AT methods. Besides, it is worth noting that we do **not** simply separate benign and adversarial samples into completely different classes (*i.e.*, respectively adopting $\mathbf{y}_i \,\|\, \mathbf{0}$ and $\mathbf{0} \,\|\, \mathbf{y}_i$ as their new supervised labels). Instead, we construct two-hot soft labels so that the separation also becomes a certain learnable pattern within the optimization process to further utilize the potential of DNNs. We demonstrate the reason for this design in Section 2.2.3.

Albeit the previous work DuRM (Wang et al., 2023a) also involves additional dummy classes, there are three essential differences between it and our work:

- **Motivation**. DuRM expects dummy classes to provide implicit supervision for original ones to facilitate standard generalization of ERM. Differently, we involve dummy classes to tolerate different distributions between benign and adversarial samples, so that the two AT objectives namely clean accuracy and robustness can no longer be at odds with each other.

- **Approach**. DuRM tends to adopt a smaller $C_d$ (*e.g.*, typically 1 or 2) than $C$, and expects there should be no samples actually classified to the dummy classes. In contrast, our paradigm always adopts another $C$ dummy classes, clearly encouraging adversarial samples to be classified into them, and utilizing the one-to-one correspondences between original and dummy classes to detect and recover adversarial samples at inference time, thus acquiring certain robustness.

- **Achievement**. Despite AT has been mentioned as one of several application scenarios of DuRM, as it actually does not involve any special designs for AT, the reported improvement is trivial (*i.e.*, no significant advancements compared with baselines like PGD-AT). On the contrary, our approach achieves SOTA performance as shown in Section 3.

### 2.2.3 A TOY CASE

Before specifically proposing our AT method under the new paradigm formulated above, we first demonstrate a generalization problem crossing different adversaries through a toy case in this section, which also double-confirms the reasonability of introducing the unique two-hot soft label in our new paradigm. Specifically, we directly assign that $\dot{\mathbf{y}}_i = \ddot{\mathbf{y}}_i' = \mathbf{y}$ and $\ddot{\mathbf{y}}_i = \dot{\mathbf{y}}_i' = \mathbf{0}$, which means there is no supervision information to explicitly bridge corresponding benign and adversarial samples, and they are viewed to belong to absolutely different classes. We implement such a toy case with the same settings as the proof-of-concept experiments in Section 2.1, except additionally introducing Auto-Attack (Croce & Hein, 2020), which is one of the most solid adaptive attack ensembles for reliable test of robust generalization, as a more powerful adversary. As shown in Figure 4, our result demonstrates that although this toy case can achieve surprisingly high accuracy both on clean data and under PGD-10 adversary, it immediately collapses once facing Auto-Attack, which reveals that it fails to learn the real robustness that

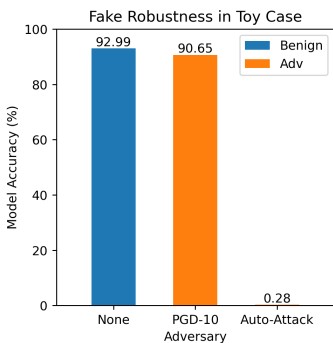

Figure 4: A toy case using hard labels achieves fake robustness.

can generalize to unseen adversaries. On the contrary, it simply overfits the PGD-10 training samples. This toy case suggests that the most straightforward idea to completely separate the learning of benign and adversarial samples may be undesirable, implying the importance of building generalized correspondence between them. This not only supports the design of two-hot soft labels in our

paradigm but also provides empirical guidance for our specific AT method proposed in the following Section 2.3. Besides, please kindly note that our two-hot soft labels are also distinguished from the conventional soft labels, such as the ones in Müller et al. (2019), Shafahi et al. (2019), and Wu et al. (2024). We defer the detailed discussion to Appendix D.1.

## 2.3 PROPOSED METHOD: DUMMY CLASSES-BASED ADVERSARIAL TRAINING

In this section, following our new AT paradigm, we specifically propose a novel AT method named **DU**mmy **C**lasses-based **A**dversarial **T**raining (DUCAT). Formally, for the target model $\boldsymbol{\theta}$ and certain benign sample $\mathbf{x}_i$, given $\hat{\mathbf{x}}'_i$ representing the adversarial sample generated through $\hat{\mathbf{x}}'_i = \arg\max_{\mathbf{x}'_i \in \mathcal{P}(\mathbf{x}_i)} \mathbb{1}(\arg\max_{k=1,\dots,C} h_{\boldsymbol{\theta}}(\mathbf{x}'_i)^{(k)} \neq y_i)$, then based on Equations (5) and (6), our new empirical risk to be minimized can be formulated with the *0-1 loss* (Zhang et al., 2019) as follows:

$$
\begin{aligned}
\mathcal{R}^{\text{DUCAT}}(\boldsymbol{\theta}, \mathbf{x}_i) := &\; \alpha \cdot \big(\beta_1 \cdot \mathbb{1}(\, g_{\boldsymbol{\theta}}(\mathbf{x}_i) \neq y_i\,) + (1-\beta_1) \cdot \mathbb{1}(\, g_{\boldsymbol{\theta}}(\mathbf{x}_i) \neq (y_i+C))\big) \\
&+ (1-\alpha) \cdot \big(\beta_2 \cdot \mathbb{1}(\, g_{\boldsymbol{\theta}}(\mathbf{x}'_i) \neq (y_i+C)\,) + (1-\beta_2) \cdot \mathbb{1}(\, g_{\boldsymbol{\theta}}(\mathbf{x}'_i) \neq y_i)\big),
\end{aligned}
\tag{7}
$$

where $\mathbb{1}(\cdot)$ represents the *indicator function* and the weights $\alpha$, $\beta_1$, $\beta_2$ are hyper-parameters.

Such a deconstruction of our target risk is expected to facilitate the understanding of its principle and novelty. Specifically, existing AT methods either only focus on the adversarial risk $\mathbb{1}(\, h_{\boldsymbol{\theta}}(\mathbf{x}'_i) \neq y_i)$ or jointly considering the benign ones like $\mathbb{1}(\, h_{\boldsymbol{\theta}}(\mathbf{x}_i) \neq y_i) + \mathbb{1}(\, h_{\boldsymbol{\theta}}(\mathbf{x}'_i) \neq y_i)$ (Bai et al., 2021). Our toy case in Section 2.2.3 attributes benign and adversarial samples to completely irrelevant risks as $\mathbb{1}(\, h_{\boldsymbol{\theta}}(\mathbf{x}_i) \neq y_i) + \mathbb{1}(\, h_{\boldsymbol{\theta}}(\mathbf{x}'_i) \neq (y_i+C))$, which is also confirmed undesirable. In contrast, our proposed risk expects the benign and adversarial samples with the same original label to be classified as an original class and a dummy one. The important difference is that **neither** the original class is necessary to be completely benign **nor** the dummy one is expected totally adversarial. The simple philosophy behind this idea is that assuming corresponding benign and adversarial samples to have completely different distributions is as unadvisable as assuming the same distributions between them. Instead, we utilize the weights $\beta_1$ and $\beta_2$ to inject our preference in a softer manner, suggesting that when learning the primary target label of a benign sample $\mathbf{x}_i$ (or adversarial sample $\mathbf{x}'_i$) is found too hard for DNNs, automatically shunting them to the corresponding dummy (or benign) class is a more acceptable alternative option compared with other completely irrelevant classes. This approach is expected to not only reduce the overfitting to specific adversaries caused by the memorization of hard-label adversarial samples in the training process (Stutz et al., 2020; Dong et al., 2022; Cheng et al., 2022), but also establish explicit connections between corresponding benign and adversarial samples, laying a solid foundation for the inference time projection from dummy classes to the corresponding original ones as provided in Equation (6).

Finally, while *0-1 loss* benefits conceptual analysis from the perspective of the risk, the optimization directly over it can be computationally intractable. So to end up with a real-world practical AT method, we introduce *CE*, the most commonly used for both supervised classification and conventional AT, as the surrogate loss to optimize the proposed risk $\mathcal{R}^{\text{DUCAT}}$ in Equation (7). It is also feasible to adopt more advanced loss functions such as *boosted CE* (Wang et al., 2020b), which creates conditions for integrating the proposed DUCAT with conventional AT methods. Formally, provided $l(\mathbf{y}_i, \beta)$ being the two-hot soft label constructed as the supervision signal, such that:

$$
l(\mathbf{y}_i, \beta) = (\beta \cdot \mathbf{y}_i) \,\|\, ((1-\beta) \cdot \mathbf{y}_i).
\tag{8}
$$

The **overall objective** of the proposed DUCAT can be formulated as:

$$
\mathcal{L}^{\text{DUCAT}}(\boldsymbol{\theta}, \mathcal{D}, \alpha, \beta_1, \beta_2) := \frac{1}{n} \sum_{i=1}^{n} \Big(\alpha \cdot \mathcal{L}^{\text{surro}}\big(\mathbf{x}_i, l(\mathbf{y}_i, \beta_1)\big) + (1-\alpha) \cdot \mathcal{L}^{\text{surro}}\big(\hat{\mathbf{x}}'_i, l(\mathbf{y}_i, (1-\beta_2))\big)\Big).
\tag{9}
$$

Then with *CE* $\mathcal{L}^{\text{CE}}$ serving as the surrogate loss $\mathcal{L}^{\text{surro}}$, we have:

$$
\begin{aligned}
\mathcal{L}^{\text{DUCAT}}(\boldsymbol{\theta}, \mathcal{D}, \alpha, \beta_1, \beta_2) = \frac{1}{n} \sum_{i=1}^{n} \Big( &-\alpha \cdot \big(\beta_1 \cdot \log(\mathbf{p}^{(y_i)}(\mathbf{x}_i, \boldsymbol{\theta})) + (1-\beta_1) \cdot \log(\mathbf{p}^{(y_i+C)}(\mathbf{x}_i, \boldsymbol{\theta}))\big) \\
&-(1-\alpha) \cdot \big(\beta_2 \cdot \log(\mathbf{p}^{(y_i+C)}(\hat{\mathbf{x}}'_i, \boldsymbol{\theta})) + (1-\beta_2) \cdot \log(\mathbf{p}^{(y_i)}(\hat{\mathbf{x}}'_i, \boldsymbol{\theta}))\big)\Big),
\end{aligned}
\tag{10}
$$

in which with $\sigma : \mathbb{R}^{2C} \to (0, 1)^{2C}$ representing *softmax*, $\mathbf{p}(\mathbf{x}_i, \boldsymbol{\theta})$ denotes the probabilistic prediction

| Dataset | Method | Original | | | | +DUCAT (Ours) | | | |
|---|---|---|---|---|---|---|---|---|---|
| | | Clean | PGD-10 | PGD-100 | Auto-Attack | Clean | PGD-10 | PGD-100 | Auto-Attack |
| CIFAR-10 | PGD-AT | 82.92 | 51.81 | 50.34 | 46.74 | **88.81** | 65.10 | 62.71 | **58.61** |
| | TRADES | **79.67** | 52.14 | 51.88 | 47.62 | 77.74 | 52.66 | 51.98 | **48.17** |
| | MART | 77.93 | 53.61 | 52.83 | 46.70 | **80.65** | 58.42 | 57.81 | **50.18** |
| | Consistency-AT | 83.42 | 53.20 | 51.68 | 47.72 | **89.51** | 66.83 | 63.80 | **57.18** |
| CIFAR-100 | PGD-AT | 56.56 | 29.27 | 28.71 | 25.02 | **70.71** | 33.17 | 29.56 | **25.20** |
| | TRADES | 55.39 | 29.61 | 29.28 | 24.51 | **55.41** | 30.69 | 30.38 | **25.23** |
| | MART | 49.83 | 30.60 | 30.31 | 25.00 | **56.73** | 41.78 | 34.32 | **27.44** |
| | Consistency-AT | 58.53 | 29.99 | 29.13 | 25.39 | **72.29** | 33.98 | 30.73 | **25.66** |
| Tiny-ImageNet | PGD-AT | 46.32 | 21.75 | 21.52 | 17.07 | **56.18** | 24.23 | 22.54 | **18.68** |
| | TRADES | 46.75 | 21.62 | 21.52 | 16.60 | **46.90** | 22.38 | 22.08 | **17.27** |
| | MART | 39.70 | 22.98 | 22.79 | 17.18 | **43.37** | 25.23 | 25.68 | **18.41** |
| | Consistency-AT | 46.54 | 22.58 | 21.70 | 17.60 | **59.64** | 25.40 | 23.22 | **18.92** |

Table 1: The results of integrating the proposed DUCAT to four AT benchmarks on CIFAR-10, CIFAR-100 and Tiny-ImageNet with ResNet-18 under $\ell_\infty$ adversaries, clearly releasing the inherent trade-off between clean accuracy and robustness. All results are acquired from three runs.

vector from DUCAT (*i.e.*, with length $2C$ and the components adding up to one) such that:

$$\mathbf{p}(\mathbf{x}_i, \boldsymbol{\theta}) = \sigma\big(h_{\boldsymbol{\theta}}(\mathbf{x}_i) \,\|\, h_{\boldsymbol{\theta}}^{\text{Dummy}}(\mathbf{x}_i)\big) = \sigma\big(\mathbf{q}(\mathbf{x}_i, \boldsymbol{\theta}) \,\|\, \mathbf{q}^{\text{Dummy}}(\mathbf{x}_i, \boldsymbol{\theta})\big). \tag{11}$$

## 3 EXPERIMENTS

Following previous works in this area, the evaluation of the proposed method is conducted on CIFAR-10, CIFAR-100 (Krizhevsky & Hinton, 2009) and Tiny-ImageNet (Li et al., 2015) datasets with the ResNet-18 (He et al., 2016) and WideResNet-28-10 (Zagoruyko & Komodakis, 2016) architectures. The three most commonly used AT baselines namely PGD-AT, TRADES, MART, and a SOTA method, Consistency-AT, are adopted as experimental benchmarks. Additionally, we involve 16 related SOTAs specifically for the trade-off problem in our comparison by simply adopting their advances reported in the original papers. These AT methods are respectively detailed in Appendix A.2 and A.3. An example algorithm of PGD-AT + DUCAT can be found in Appendix B.

### 3.1 EXPERIMENTAL SETUP

**Training details.** We adopt SGD optimizer with momentum 0.9; batch size 128; weight decay $5 \times 10^{-4}$; initial learning rate 0.1; total 130 training epochs with learning rate decay by a factor of 0.1 at 100 and 105 epochs, respectively. Except for extending the training epoch which is originally 110 for better convergence, we follow the common settings as suggested by Pang et al. (2021), which studies various tricks and hyper-parameters of AT. Standard data pre-processing is also involved in the training process, with normalization of all natural images into [0, 1] and data augmentation including random crop with 4-pixel zero padding and 50% random horizontal flip. Then regarding the hyper-parameters specific to different AT methods, for the proposed DUCAT, we uniformly assign $\alpha = 0.5$, $\beta_1 = 0.75$ and $\beta_2 = 1$, respectively starting at epoch 105 for CIFAR-10 and 100 for the other two datasets. And for the benchmarks, we adopt the same settings as their original papers. The experiments are conducted on Ubuntu 22.04 with 512GB RAM and $8 \times$ NVIDIA GeForce RTX 4090 GPUs, and are implemented with Python 3.8.19 and PyTorch 1.8.1+cu111.

**Threat models.** In the main experiments, we consider untargeted white-box attacks under $\ell_\infty$ norm, with the maximal perturbation budget $\epsilon = 8/255$ and optimization step size $\alpha = 2/255$, which is the most common threat model for examining the adversarial robustness. We additionally consider other categories of threat models in Appendix C.1, including the one adopting the targeted attack. For adversary capacity, we assume full model information is available, so it is also visible to the adversary that the DNN under DUCAT defense will output logits with length $2C$. However, the runtime projection is not a part of the computational graph of the model, and the specific correspondences between benign and dummy classes are also diverse and underlying. Specifically, it is not

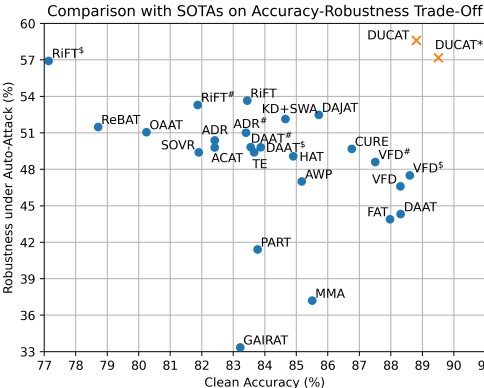

Figure 5: Comparison between the proposed DUCAT and 16 current SOTAs in the trade-off problem on CIFAR-10 and ResNet-18. The superscripts #, $, and * in the figure refer to the integration with TRADES, MART and Consistency-AT, respectively. Here we highlight the DUCAT and DUCAT* as they are the ones namely achieve the best adversarial robustness and clean accuracy among our solutions under this comparison settings. The result demonstrates the significant advancement contributed by our work.

necessary to implement them in order such that class $k + C$ is exactly the dummy class of class $k$. In that case, the adversary is not able to know the one-to-one correspondences unless inferring them by multiple querying as in black-box attacks.

## 3.2 MAIN RESULTS

**Effectiveness in improving benchmarks.** As a plug-and-play method, DUCAT can be easily integrated into the four experimental benchmarks. In this section, we demonstrate the improvement brought by such integration to confirm the wide effectiveness of our new paradigm and method. Specifically, we report both clean accuracy on benign test samples and adversarial robustness under three adversaries, namely PGD-10, PGD-100 (Madry et al., 2018) and Auto-Attack (Croce & Hein, 2020) with default settings and random start. All the reported results are averages of three runs, with the specific performance of each run acquired on the best checkpoint achieving the highest PGD-10 accuracy. The results on ResNet-18 in Table 1 show that DUCAT significantly improves all benchmarks in both accuracy and robustness across all datasets, demonstrating its general effectiveness and confirming the success of our new paradigm in releasing the current trade-off between clean and adversarial targets. Additional results on WideResNet-28-10 are provided by Table 3 in Appendix C, and a discussion about the relatively less improvement of TRADES + DUCAT is in Appendix D.2.

**Compared with 16 SOTAs.** To better show the advancement contributed by our work, we additionally compare with the SOTA related works that directly aim at releasing the trade-off between accuracy and robustness. Specific for CIFAR-10 and ResNet-18 without extra training data, we investigate the advances in the trade-off problem over the past five years, including (Ding et al., 2020; Wu et al., 2020; Zhang et al., 2020; 2021; Chen et al., 2021; Addepalli et al., 2022b;a; Dong et al., 2022; Rade & Moosavi-Dezfooli, 2022; Yang & Xu, 2022; Kanai et al., 2023; Zhu et al., 2023; Cao et al., 2024; Gowda et al., 2024; Wu et al., 2024; Zhang et al., 2024; Wang et al., 2024b), and make a comparison with the results reported in their original papers. Considering the fairness, for works without original results under such settings, we turn to reliable external sources including the *RobustBench* leaderboard (Croce et al., 2021) and other published papers such as Liu et al. (2021). The comparison result is shown in Figure 5, with more details deferred to Table 2 in Appendix A.3.

**Low additional cost in efficiency.** As DUCAT doubles the last fully connected layer, the model complexity can accordingly increase, which indeed brings an additional price in the training efficiency. However, compared with the parameters of the whole model, the new parameters introduced are expected to be minor, and as a result, would not significantly degrade the efficiency. As shown in Figure 6, the comparison in time cost before and after integrating DUCAT into the four experimental benchmarks supports this statement, among which the maximum additional time cost is still less than 20%. This means the additional implementation of DUCAT can be lightweight, which also fits our vision of a more practical AT for real-world applications.

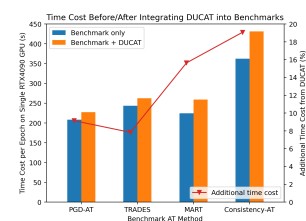

Figure 6: DUCAT introduces not much additional time cost.

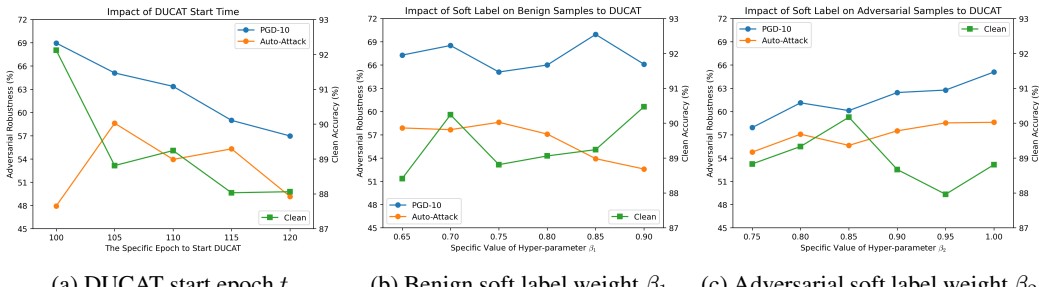

(a) DUCAT start epoch $t$.      (b) Benign soft label weight $\beta_1$.      (c) Adversarial soft label weight $\beta_2$.

Figure 7: Comprehensive ablation studies of the proposed DUCAT on CIFAR-10 and ResNet-18 *w.r.t.* three hyper-parameters, namely $t$, $\beta_1$ and $\beta_2$. The y-axises are aligned across the subfigures.

## 3.3 ABLATION STUDY

Provided the practical simplicity of the proposed DUCAT, there are mainly four hyper-parameters that impact the trade-off between accuracy and robustness for different reasons, namely the specific epoch $t$ to start DUCAT, and the $\alpha$, $\beta_1$, $\beta_2$ in Equation (7). As $\alpha$ directly adjusts the learning preference, attaching more importance to either benign or adversarial samples, which has a predictable impact (*i.e.*, either better accuracy or robustness) and is not specific to our method (*e.g.*, the regularization parameter $\lambda$ in TRADES is based on a similar idea), we defer its study to Appendix C.2. While in this section, on CIFAR-10 and ResNet-18, we conduct ablation studies for the start epoch $t$ (Figure 7a), as well as the two-hot soft label construction weight $\beta_1$ (Figure 7b) and $\beta_2$ (Figure 7c) respectively for benign and adversarial samples, on the trade-off between clean accuracy (*i.e.*, right y-axis and green line) and adversarial robustness (*i.e.*, left y-axis, along with blue and orange line respectively for PGD-10 and Auto-Attack).

The first important phenomenon observed is that an appropriate epoch $t$ to start integrating DUCAT matters. Specifically, when starting too early (*e.g.*, $t = 100$), the overfitting to the training adversary PGD-10 can be serious, resulting in up to 20% robust generalization gap to test-time Auto-Attack. Yet, if starting too late (*e.g.*, $t = 120$), the learning can be insufficient in the first place, with both accuracy and robustness unsatisfying. This also reveals an important potential of DUCAT to serve as an adversarial **fine-turning** technology that can swiftly enhance already implemented robust DNN without retraining them from scratch. More specifically, provided that the existing target model is originally built through conventional AT by an appropriate epoch (*e.g.*, epoch 100), we can resume the training from that epoch with DUCAT integrated. Then after a few epochs (*e.g.*, before epoch 130), the model can achieve a better trade-off between clean accuracy and robustness beating existing SOTAs. This makes DUCAT also valuable in updating real-world robust applications.

For another, though as expected, the accuracy (or robustness) increases while robustness (or accuracy) drops as $\beta_1$ (or $\beta_2$) approaches 1, as long as within a reasonable range such that the original and dummy labels respectively serve as the primary learning target of benign and adversarial samples, the specific weight $\beta_1$ and $\beta_2$ to construct two-hot soft labels mildly impact DUCAT performance, demonstrating its stability. Interestingly, the optimal range of $\beta_2$ (*e.g.*, [0.9, 1]) is larger in value than $\beta_1$ (*e.g.*, [0.7, 0.8]), which is probably because learning adversarial samples is still harder than benign ones after being separated as different objectives, thus need to ensure more training samples.

## 4 CONCLUSION

In this work, given the inherent trade-off between clean accuracy and robustness in AT and the undesirable recent progress on this problem, we reveal that always-failed samples widely exist in conventional AT, explicitly attributing such a trade-off to the overstrict assumption of the current AT paradigm. In response, we suggest a new AT paradigm with dummy classes to relax this assumption, and accordingly propose a plug-and-play DUCAT method, releasing the trade-off and outperforming four benchmarks plus 16 SOTAs. For future works, more advanced methods under the new AT paradigm might be a promising direction, and we hope this work could attract more attention and inspire further studies on different paths from the current one to both accurate and robust DNNs.

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

APPENDIX

# A    DETAILS OF ADVERSARIES AND AT METHODS INVOLVED

In this section, we supplement the introduction of the adversarial attack technologies serving as adversaries in this work, including PGD and Auto-Attack. Also, we introduce more details about the AT methods involved, including four benchmarks namely PGD-AT (Madry et al., 2018), TRADES (Zhang et al., 2019), MART (Wang et al., 2020b) and Consistency-AT (Tack et al., 2022), as well as 16 SOTAs on the trade-off problem, respectively AWP (Wu et al., 2020), FAT (Zhang et al., 2020), MMA (Ding et al., 2020), GAIRAT (Zhang et al., 2021), KD + SWA (Chen et al., 2021), ACAT (Addepalli et al., 2022b), DAAT (Yang & Xu, 2022), DAJAT (Addepalli et al., 2022b), HAT (Rade & Moosavi-Dezfooli, 2022), OAAT (Addepalli et al., 2022a), TE (Dong et al., 2022), RiFT (Zhu et al., 2023), SOVR (Kanai et al., 2023), ADR (Wu et al., 2024), CURE (Gowda et al., 2024), PART (Zhang et al., 2024), VFD (Cao et al., 2024) and ReBAT (Wang et al., 2024b).

## A.1    ADVERSARIAL ROBUSTNESS AND ATTACK TECHNOLOGIES

Adversarial robustness refers to the performance of DNN classifiers under malicious perturbations by any possible adversaries, which is commonly measured with the test accuracy under adversarial attacks (Bai et al., 2021). The concept of adversarial attack is proposed by Biggio et al. (2013) and Szegedy et al. (2014), followed by a basic write-box method, *fast gradient sign method* (FGSM) (Goodfellow et al., 2015), that generates the perturbation based on the gradient of the loss function *w.r.t.* the input sample. The *projected gradient descent attack* (PGD) (Madry et al., 2018) can be viewed as a variant of FGSM, which first produces the perturbation by iteratively running FGSM, and then projects it back into the $\epsilon$-ball of the input sample. Nowadays, as Auto-Attack (Croce & Hein, 2020) forms a parameter-free and user-independent ensemble of attacks, which can identify frequent pitfalls in AT practice including over-adjustment of hyper-parameters and gradient obfuscation or masking, it has been widely recognized as one of the most reliable adversaries to exam robustness.

## A.2    EXPERIMENTAL AT BENCHMARKS

**PGD-AT** (Madry et al., 2018).    This is the first method demonstrated to be effective for solving the *min-max* problem of the current AT paradigm in Equation (2) and training moderately robust DNNs (Athalye et al., 2018; Wang et al., 2020b). To be specific, based on *Danskin's Theorem* (Danskin, 2012), PGD-AT proposed to find a constrained maximizer of the inner maximization by PGD, which is believed sufficiently close to the optimal attack, and then use the maximizer as an actual data point for the outer minimization through gradient descent.

**TRADES** (Zhang et al., 2019).    As another typical AT method, TRADES proposed to decompose the robust error into natural error and boundary error, thus directly balancing the trade-off between natural accuracy and robustness. Specifically, boundary error occurs when the specific data point is sufficiently close to the decision boundary that can easily cross it under slight perturbation, which is believed as one reason for the existence of adversarial samples (Bai et al., 2021; Wang et al., 2024a).

**MART** (Wang et al., 2020b).    A problem of TRADES is that the boundary error is designed to push each pair of benign and adversarial samples together, no matter whether the benign data are classified correctly or not (Bai et al., 2021). As a follow-up work, MART further investigates the influence of correctly classified and misclassified samples for adversarial robustness separately, and then suggests adopting additional boundary error *w.r.t.* misclassified samples.

**Consistency-AT** (Tack et al., 2022).    It is found that *consistency regularization* forces the model to give the same output distribution when the input or weights are slightly perturbed, which fits the goal of AT when the perturbation is generated adversarially (Zhang et al., 2022). Consistency-AT proposed a new target that the predictive distributions after attacking from two different augmentations of the same instance should be similar to each other. The underlying principle is that adversarial robustness essentially refers to model stability around naturally occurring inputs, learning to satisfy such a constraint should not inherently require labels (Carmon et al., 2019), which also relaxes the current assumption from a different perspective.

### A.3 SOTA AT METHODS ON TRADE-OFF BETWEEN ACCURACY AND ROBUSTNESS

The trade-off between clean accuracy and adversarial robustness has been explored from various perspectives under the current AT paradigm. Below we list 16 representative SOTAs on this problem in the past five years, which are adopted to demonstrate the advancement achieved by our new AT paradigm and DUCAT method in Section 3.2. In this section, we show the sources and detailed records for these SOTAs in Table 2, which are the corresponding raw data we rely on to draw the aforementioned Figure 5. Besides, we also briefly introduce the principle for each of these SOTAs, except for FAT (Zhang et al., 2020), HAT (Rade & Moosavi-Dezfooli, 2022) and SOVR (Kanai et al., 2023) that have been introduced in Section 1.

| Year | Method | Clean | Auto-Attack | Record Source |
|------|--------|-------|-------------|---------------|
| 2020 | AWP (Wu et al., 2020) | 85.17 | 47.00 | Liu et al. (2021) |
|      | FAT (Zhang et al., 2020) | 87.97 | 43.90 | Liu et al. (2021) |
|      | MMA (Ding et al., 2020) | 85.50 | 37.20 | original paper |
| 2021 | GAIRAT (Zhang et al., 2021) | 83.22 | 33.35 | Liu et al. (2021) |
|      | KD + SWA (Chen et al., 2021) | 84.65 | 52.14 | original paper |
| 2022 | ACAT (Addepalli et al., 2022b) | 82.41 | 49.80 | original paper |
|      | DAAT (Yang & Xu, 2022) | 88.31 | 44.32 | original paper |
|      | DAJAT (Addepalli et al., 2022b) | 85.71 | 52.48 | *RobustBench* |
|      | HAT (Rade & Moosavi-Dezfooli, 2022) | 84.90 | 49.08 | original paper |
|      | OAAT (Addepalli et al., 2022a) | 80.24 | 51.06 | *RobustBench* |
|      | TE (Dong et al., 2022) | 83.66 | 49.40 | original paper |
| 2023 | RiFT (Zhu et al., 2023) | 83.44 | 53.65 | original paper |
|      | SOVR (Kanai et al., 2023) | 81.90 | 49.40 | original paper |
| 2024 | ADR (Wu et al., 2024) | 82.41 | 50.38 | original paper |
|      | CURE (Gowda et al., 2024) | 86.76 | 49.69 | original paper |
|      | PART (Zhang et al., 2024) | 83.77 | 41.41 | original paper |
|      | VFD (Cao et al., 2024) | 88.30 | 46.60 | original paper |
|      | ReBAT (Wang et al., 2024b) | 78.71 | 51.49 | original paper |
|      | **Ours (PGD-AT + DUCAT)** | **88.81** | **58.61** | - |
|      | **Ours (Consistency-AT + DUCAT)** | **89.51** | **57.18** | - |

Table 2: Corresponding to Figure 5 in Section 3.2, this table illustrates the raw data of the 16 SOTAs on the trade-off problem, demonstrating the advancement of our work. Note that ReBAT is the only one here with PreActResNet-18 due to the lack of ResNet-18 results in its original paper.

AWP (Wu et al., 2020) proposed a double perturbation mechanism that can flatten the loss landscape by weight perturbation to improve robust generalization. MMA (Ding et al., 2020) proposed to use adaptive $\epsilon$ for adversarial perturbations to directly estimate and maximize the margin between data and the decision boundary. GAIRAT (Zhang et al., 2021) proposed that a natural data point closer to (or farther from) the class boundary is less (or more) robust, and the corresponding adversarial data point should be assigned with larger (or smaller) weight. DAJAT (Addepalli et al., 2022b) aims to handle the conflicting goals of enhancing the diversity of the training dataset and training with data that is close to the test distribution by using a combination of simple and complex augmentations with separate batch normalization layers. ACAT (Addepalli et al., 2022b) is a two-step variant of DAJAT to improve computational efficiency. OAAT (Addepalli et al., 2022a) aligns the predictions of the model with that of an Oracle during AT to achieve robustness within larger bounds.

Enlightening, TE (Dong et al., 2022) proposed that one-hot labels can be noisy for them because they naturally lie close to the decision boundary, which makes it essentially difficult to assign high-confident one-hot labels for all perturbed samples within the $\epsilon$-ball of them (Stutz et al., 2020; Cheng et al., 2022). So the model may try to memorize these hard samples during AT, resulting in robust overfitting. This may also be another reason why AT leads to more complicated decision boundary, and explains why robust overfitting is harder to alleviate than the clean one. Based on a similar idea, KD + SWA (Chen et al., 2021) investigates two empirical means to inject more learned smoothening during AT, namely leveraging knowledge distillation and self-training to smooth the logits, as well

| Dataset | Method | Original | | | | +DUCAT (Ours) | | | |
|---|---|---|---|---|---|---|---|---|---|
| | | Clean | PGD-10 | PGD-100 | Auto-Attack | Clean | PGD-10 | PGD-100 | Auto-Attack |
| CIFAR-10 | PGD-AT | 87.49 | 55.81 | 54.37 | 50.96 | **90.02** | 64.53 | 62.76 | **58.77** |
| | TRADES | **85.35** | 57.15 | 56.17 | 51.88 | 83.62 | 57.77 | 57.00 | **53.78** |
| | MART | 82.78 | 58.47 | 57.47 | 50.89 | **84.64** | 58.26 | 57.07 | **54.64** |
| | Consistency-AT | 86.90 | 55.71 | 54.41 | 50.83 | **89.86** | 67.89 | 64.17 | **59.44** |
| CIFAR-100 | PGD-AT | 59.95 | 32.18 | 31.21 | 27.27 | **70.02** | 34.28 | 30.73 | **27.95** |
| | TRADES | 59.51 | 32.51 | 32.34 | 27.71 | **59.70** | 33.77 | 33.38 | **28.55** |
| | MART | 56.84 | 34.12 | 33.70 | 27.97 | **60.97** | 36.80 | 35.98 | **32.40** |
| | Consistency-AT | 60.84 | 32.48 | 31.64 | 27.74 | **72.97** | 35.90 | 32.24 | **28.31** |
| Tiny-ImageNet | PGD-AT | 47.79 | 23.97 | 23.59 | **20.00** | 62.58 | 25.86 | 23.68 | 18.65 |
| | TRADES | 51.14 | 24.84 | 24.58 | 20.02 | **51.91** | 25.77 | 25.39 | **20.22** |
| | MART | 45.57 | 26.21 | 25.92 | 21.07 | **50.71** | 28.48 | 28.64 | **26.15** |
| | Consistency-AT | 50.12 | 25.05 | 24.42 | 20.64 | **62.77** | 25.62 | 25.01 | **22.09** |

Table 3: The additional results of integrating the proposed DUCAT to four AT benchmarks on CIFAR-10, CIFAR-100 and Tiny-ImageNet with WideResNet-28-10 under $\ell_\infty$ adversaries, corresponding to Table 1 in Section 3.2. All results are acquired from three runs.

as performing stochastic weight averaging (Izmailov et al., 2018) to smooth the weights, and DAAT (Yang & Xu, 2022) adaptively adjusts the perturbation ball to a proper size for each of the natural examples with the help of a naturally trained calibration network.

More recently, RiFT (Zhu et al., 2023) introduces module robust criticality, a measure that evaluates the significance of a given module to model robustness under worst-case weight perturbations, to exploit the redundant capacity for robustness by fine-tuning the adversarially trained model on its non-robust-critical module. ADR (Wu et al., 2024) generates soft labels as a better guidance mechanism that accurately reflects the distribution shift under attack during AT. CURE (Gowda et al., 2024) finds that selectively updating specific layers while preserving others can substantially enhance the network's learning capacity, and accordingly leverages a gradient prominence criterion to perform selective conservation, updating, and revision of weights. PART (Zhang et al., 2024) partially reduces $\epsilon$ for

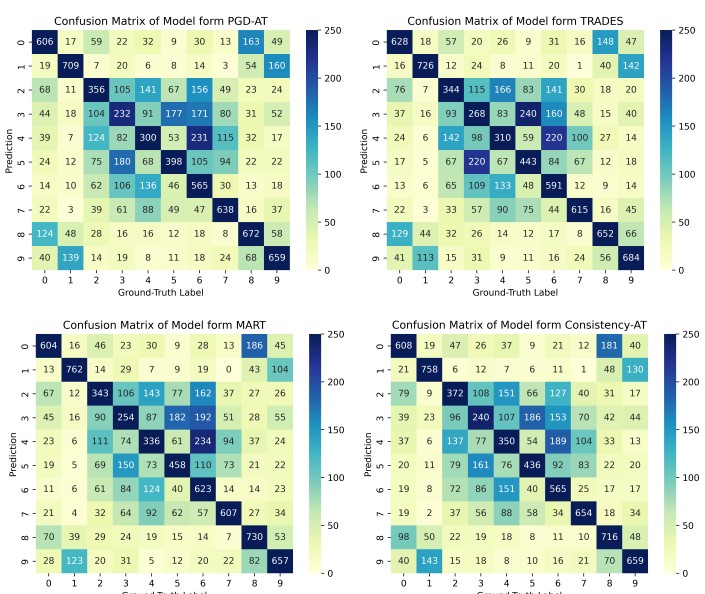

Figure 8: Corresponding to Figure 2 in Section 2.1, under the PGD-10 adversary, similar failure patterns can be observed in the confusion matrixes respectively with the protection of the four experimental AT methods.

less influential pixels, guiding the model to focus more on key regions that affect its outputs. VFD (Cao et al., 2024) conducts knowledge distillation from a pre-trained model optimized towards high accuracy to guide the AT model towards generating high-quality and well-separable features by constraining the obtained features of natural and adversarial examples. ReBAT (Wang et al., 2024b) views AT as a dynamic mini-max game between the model trainer and the attacker, and proposes to alleviate robust overfitting by rebalancing the two players by either regularizing the trainer's capacity or improving the attack strength.

## B  DUCAT ALGORITHM

In this section, we provide the specific algorithm to facilitate the understanding of the proposed DUCAT. This algorithm demonstrates the basic format of DUCAT (*i.e.*, integrated with PGD-AT), and focuses on its novel parts, thus leaving out some general training details such as optimizer settings, training adversary, mini-batch and learning rate decay, which have been detailed introduced in Section 3.1. Notice that as suggested in Section 3.3, DUCAT can be used for both training a randomly initialized model from scratch and fine-turning an already adversarially trained model for a better trade-off between accuracy and robustness, so we introduce the resuming epoch $T_r$ to enable customization in the latter case.

---

**Algorithm 1 DU**mmy **C**lasses-based **A**dversarial **T**raining (DUCAT)

---

**Input**: training dataset $\mathcal{D} = \{(\mathbf{x}_i, y_i)\}_{i=1,\dots,n}$ of a $C$-class classification task, target model $\boldsymbol{\theta}$

**General Training Parameters**: training epoch $T$, resuming epoch $T_r$, training adversary $\mathcal{A}$, loss function $\mathcal{L}$, learning rate $r$

**DUCAT hyper-parameters**: start epoch $t$, preference weight $\alpha$, benign weight $\beta_1$ and adversarial weight $\beta_2$ for two-hot soft label construction

**Output**: robust $\boldsymbol{\theta}'$ with output *logits* of length $2C$

1: Initialize $t_{curr} \leftarrow T_r$ **if provided** $T_r$ **else** $0$
2: $\boldsymbol{\theta}' \leftarrow doubleLastLayer(\boldsymbol{\theta})$
3: **while** $t_{curr} < T$ **do**
4:     Initialize $\mathcal{D}_{curr} \leftarrow [], \mathcal{D}_{benign} \leftarrow [], \mathcal{D}_{adv} \leftarrow [], \mathcal{A}_{curr} \leftarrow \mathcal{A}(\boldsymbol{\theta}')$
5:     **while** $\mathbf{x}_i, y_i \in \mathcal{D}$ **do**
6:         $\mathbf{x}'_i \leftarrow \mathcal{A}_{curr}(\mathbf{x}_i)$
7:         **if** $t_{curr} \geq t$ **then**
8:             $\mathbf{y}_i \leftarrow onehot(y_i)$
9:             $\mathcal{D}_{benign}.append(\mathbf{x}_i, (\beta_1 \cdot \mathbf{y}_i) \| ((1 - \beta_1) \cdot \mathbf{y}_i))$
10:            $\mathcal{D}_{adv}.append(\mathbf{x}'_i, ((1 - \beta_2) \cdot \mathbf{y}_i) \| (\beta_2 \cdot \mathbf{y}_i))$
11:        **else**
12:            $\mathcal{D}_{curr}.append(\mathbf{x}'_i, y_i)$
13:        **end if**
14:    **end while**
15:    **if** $t_{curr} \geq t$ **then**
16:        $\boldsymbol{\theta}' \leftarrow \boldsymbol{\theta}' - r \cdot [\alpha \cdot \nabla_{\boldsymbol{\theta}'} \mathcal{L}(\mathcal{D}_{benign}) + (1 - \alpha) \cdot \nabla_{\boldsymbol{\theta}'} \mathcal{L}(\mathcal{D}_{adv})]$
17:    **else**
18:        $\boldsymbol{\theta}' \leftarrow \boldsymbol{\theta}' - r \cdot \nabla_{\boldsymbol{\theta}'} \mathcal{L}(\mathcal{D}_{curr})$
19:    **end if**
20:    $t_{curr} \leftarrow t_{curr} + 1$
21: **end while**
22: **return** $\boldsymbol{\theta}'$

---

## C  SUPPLEMENTARY EXPERIMENTS

In this section, we supplement more experimental results to further evaluate this work and support our contributions. Specifically, except Figure 8 and Table 3 that are directly referred to in the main body, here we also provide generalization analysis under a different threat model with targeted attacks, as well as an additional ablation study for the hyper-parameter $\alpha$. Besides, we provided additional comparisons with synthetic data-based AT methods, as well as an inference time method that is also to acquire adversarial robustness beyond the conventional AT paradigm.

### C.1  GENERALIZATION TO TARGETED THREAT MODEL

Someone may wonder, as the DUCAT defense is visible to white-box adversaries through the double-size last layer, whether they can do something to bypass this defense. One idea that might be representative is, given that DUCAT benefits from "inducing" adversaries to perturb benign samples to the one-to-one corresponding dummy classes (*i.e.*, in other words, each dummy class serves as the

| Method | Original | | +DUCAT (Ours) | | |
|---|---|---|---|---|---|
| | Untargeted | Targeted | Untargeted | Targeted (original) | Targeted (dummy) |
| PGD-AT | 51.81 | 73.02 | 65.10 | 70.23 | 70.77 |
| TRADES | 52.14 | 72.68 | 52.66 | 73.63 | 73.32 |
| MART | 53.61 | 70.48 | 58.42 | 73.87 | 72.89 |
| Consistency-AT | 53.20 | 74.26 | 66.83 | 72.41 | 72.80 |

Table 4: Additional results on targeted PGD-10 adversary compared with the default untargeted one. The "original"/"dummy" suggests that the target classes are randomly selected from the original/dummy classes. All results are acquired from three runs.

easiest attack target for the benign samples from the corresponding original class) to run-time detect and recover them, if it is possible to change from untargeted adversaries to targeted ones, so that to release such one-to-one correspondences by compulsively appointing other attack targets. Unfortunately, in this part, we demonstrate that no matter randomly appointing different original classes or dummy ones as the attack targets, the targeted PGD-10 adversary is still not sufficiently dangerous under the DUCAT defense. As the results shown in Table 4, probably due to the original difficulty of targeted attacks compared to untargeted ones, the effectiveness of targeted PGD-10 observed here is even worse than untargeted PGD-10. However, we acknowledge that such difference in effectiveness is indeed smaller in DUCAT (*i.e.*, 11.74%) than in the original benchmarks (*i.e.*, 19.92%) on average, which somewhat implies this direction might be still promising to degrade DUCAT suppose that more advanced approaches can be further proposed.

## C.2 Additional Ablation Study

Hyper-parameter $\alpha$ is designed to adjust the weights of benign and adversarial losses for the model update procedure, thus directly injecting user preference on either clean accuracy or adversarial robustness. In this work, we do not suggest fine-turning $\alpha$ as an approach to impact the trade-off, because our idea implies equal importance of benign and adversarial samples respectively regarding the learning of accuracy and robustness, and our unique two-hot soft label construction has already considered an appropriate balance between them. However, as some previous works on the trade-off problem adopt a similar hyper-parameter (*e.g.*, the regularization parameter $\lambda$ in TRADES), we still provide such an option in case our user does have any specific requirements on it in their practices. Predictably, as illustrated in Figure 9, as $\alpha$ increases, which means more importance is attached to the benign samples, there is overall an increasing (or decreasing) trend for the clean accuracy (or robustness) of the model. Nevertheless, it is also worth noting that $\alpha = 0.5$ as our default one, which means no specific preference on accuracy or robustness, is basically the optimal choice, just as we expected.

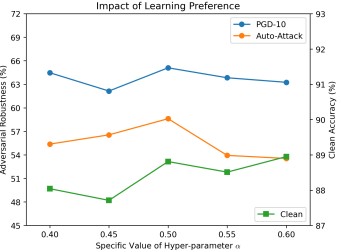

Figure 9: Ablation Study on $\alpha$ suggests that specific preference on either accuracy or robustness is undesirable for this work.

## C.3 Comparison with Synthetic Data-Based Methods

In the main experiments, we exclude related works with synthetic data, such as Rebuffi et al. (2021), Sehwag et al. (2022) and Wang et al. (2023b), because we think 1) there are certain concerns about their fairness compared with conventional AT, and 2) they focus more on data rather than algorithms, thus are not in direct competition with AT algorithms but can be easily integrated into them.

For fairness, although different from those extra data-based methods, synthetic data-based ones seem to fairly use the same training dataset as conventional AT, the problem is the dramatic additional computational cost they rely on. For instance, they need to train a diffusion model first (Rebuffi et al., 2021; Sehwag et al., 2022; Wang et al., 2023b), which is even much more time-consuming than training the robust model itself (e.g., training a ResNet-18 with PGD-AT on CIFAR-10 typically needs only about 4 hours with RTX4090, while training a DDPM (a common types of diffusion) model on CIFAR-10 usually needs up to 30 hours with A100). Although it is argued that both

training generative models and sampling from them is a one-time cost (Sehwag et al., 2022), that is only true for experimental scenarios with standard datasets, yet certainly not for specific real-world practices. Also, synthetic data-based methods rely on extra standard-trained models for pseudo-labeling of the generated unlabeled data (Rebuffi et al., 2021; Sehwag et al., 2022; Wang et al., 2023b). Besides, their training epochs and batch size are significantly larger than typical AT (Rebuffi et al., 2021; Sehwag et al., 2022; Wang et al., 2023b), which respectively means the need for more training time and the VRAM resource. We should always take these costs into consideration when adopting synthetic data-based methods.

Sometimes, a more important philosophy behind a fairness assumption is that, methods following the assumption would not be directly in competition with the ones not, and here is such a case. While conventional AT methods focus more on the advancement in algorithms, synthetic data-based methods contribute more to the data aspect. As a result, these two kinds of works would not degrade the contributions of each other but could be integrated easily. Actually, Rebuffi et al. (2021), Sehwag et al. (2022) and Wang et al. (2023b) all involve conventional AT. More

| Method | Clean | Auto-Attack |
|---|---|---|
| Wang et al. (2023b) | 91.12 | 63.35 |
| **Ours (PGD-AT + DUCAT)** | **92.60** | **62.74** |

Table 5: DUCAT with the open-source 1M CIFAR-10 synthetic data from Wang et al. (2023b) can achieve better clean accuracy and competitive robustness compared to it under significantly lighter training settings.

specifically, all of them are essentially TRADES plus different data augmentation approaches with diffusion models. In Table 5, we preliminarily showcase that by simply using the open-source 1M CIFAR-10 synthetic data from Wang et al. (2023b) without other tricks it adopts (e.g., weight averaging and cyclic learning rate schedule) and the more training epochs it needs (i.e., at least 400 epochs), we can easily train a WideResNet-28-10 achieving better clean accuracy and competitive robust performance compared with Wang et al. (2023b), by DUCAT with just the same training details of our main experiments. Considering the simplicity and computational-friendliness of DUCAT, this preliminary result is itself sufficiently considerable, while there are more promising directions like directly integrating DUCAT into synthetic data-based methods to be further explored.

## C.4 COMPARISON WITH ANOTHER NOVEL IDEA BEYOND CONVENTIONAL AT

As DUCAT suggests a novel paradigm to acquire adversarial robustness beyond conventional AT, there are also a few relevant previous works from this perspective. Representatively, Pang et al. (2020) develops Mixup Inference (MI) to mixup input with random clean samples at inference time, thus shrinking and transferring the equivalent perturbation if the input is adversarial. Although as an inference time method, MI does not directly compete with DUCAT, it is also a notable attempt at new possible directions for adversarial robustness.

| Method | Clean | PGD-10 |
|---|---|---|
| MI (Pang et al., 2020) | 84.20 | 64.50 |
| **Ours (PGD-AT + DUCAT)** | **88.81** | **65.10** |

Table 6: Comparison with another work aiming at acquiring adversarial robustness beyond the conventional AT further showcases the advantage of our DUCAT.

Thus, we additionally provide a performance comparison between DUCAT and MI on the CIFAR-10 dataset. Specifically, we follow the same settings of our main experiments for DUCAT, while adopt the performance originally reported in Pang et al. (2020) for MI. As the robust performance of MI is not evaluated by Auto-Attack there, for fairness, we compare robust accuracy under the PGD-10 adversary below, as both the two works involve it. The comparison results are shown in Table 6, where DUCAT outperforms MI though given that the results of DUCAT are acquired from ResNet-18 while those of MI are from more powerful ResNet-50.

## D ADDITIONAL DISCUSSION

### D.1 DIFFERENCE BETWEEN TWO-HOT SOFT LABEL AND CONVENTIONAL ONES

Careful readers may notice that the suggested two-hot soft label is distinguished from the conventional soft label, such as the ones proposed for label smoothing (Müller et al., 2019; Shafahi et al., 2019; Wu et al., 2024), in both motivation and specific implementation. In short, previous soft label

technologies convert one-hot label vectors into one-warm vectors that represent a low-confidence classification (Shafahi et al., 2019). In contrast, the two-hot soft label for DUCAT is neither built in a one-warm format nor aiming at a low-confidence classification. More specifically, from the perspective of motivation, our two-hot soft label is to explicitly bridge corresponding original and dummy classes as the suboptimal alternative target of each other, so that their separation also becomes learnable, while the conventional ones aim to promote learning from the soft target as better guidance that reflects the underlying distribution of data (Müller et al., 2019; Shafahi et al., 2019; Wu et al., 2024). On the other hand, from the perspective of the specific implementation, different from conventional soft label combining one-hot target with uniform or crafted distribution (Müller et al., 2019; Shafahi et al., 2019; Wu et al., 2024), our two-hot soft label just combines two one-hot targets namely as the primary and alternative targets. All in all, the unique two-hot soft label is proposed for the first time, strongly supporting the outstanding effectiveness of DUCAT.

## D.2 ANALYSIS ABOUT LESS IMPROVEMENT OF TRADES+DUCAT THAN OTHERS

Although DUCAT as an independent AT method (referred to as PGD-AT + DUCAT) achieves SOTA performance, it helps relatively less in enhancing TRADES than other AT benchmarks when serving as a plug (*i.e.*, the improvement of TRADES + DUCAT over TRADES is not impressive as others). In our opinion, this is due to the particular training adversarial samples used by TRADES, which is less appropriate for building dummy clusters than typical ones. Specifically, different from most AT methods including the other three benchmarks, PGD-AT, MART and Consistency-AT, that by default generate training adversarial samples by PGD-10, TRADES particularly crafts training adversarial samples through maximizing its own KL-divergence regularization term (Zhang et al., 2019). Because TRADES defines a boundary error measured through this KL term, which occurs when specific data points are sufficiently close to the decision boundary that can easily cross it under slight perturbation (Zhang et al., 2019), thus it makes sense to accordingly craft training adversarial samples to reduce this error. But as a consequence, these training adversarial samples naturally focus more on boundary establishment instead of reasonable data distribution patterns, which predictably degrades our DUCAT, as DUCAT assumes most of the adversarial samples from one class should belong to the corresponding dummy class.

## E LIMITATIONS

Despite the significant advancement achieved by this work, here we identify two limitations respectively for our new AT paradigm and the proposed DUCAT method, to, hopefully, facilitate future works in this area. Firstly, although the conventional trade-off between clean accuracy and robustness in the current AT paradigm has been released by our new AT paradigm, with both of them achieving new SOTA performance, we can still observe certain tension between these two objectives, though much slighter than the conventional one. It can be found in the ablation studies illustrated by Figures 7 and 9 that the trend of clean and robust curves have still not fully aligned with each other in certain ranges. This implies that our new AT paradigm does still not harmonize them perfectly, thus just serving as a stepping stone instead of an end-all solution. Also, while the clean accuracy under DUCAT is significantly improved compared with the current AT methods, there is still a gap compared with standard training (*e.g.*, the SOTA accuracy of ResNet-18 on CIFAR-10 in the clean context is about 95%). Secondly, integrating DUCAT with different benchmarks shows different degrees of advancement. As discussed in Section D.2, compared with the remarkable results with PGD-AT and Consistency-AT, as well as the considerable one with MART, the outcomes on TRADES seem less impressive. This implies that, despite the effectiveness and simplicity of the proposed DUCAT, there should be room for better specific methods under our new AT paradigm to outperform it, especially regarding the different mechanisms of specific benchmarks like TRADES.

## F   TEMPORARY FOR REBUTTAL (TO BE REMOVED)

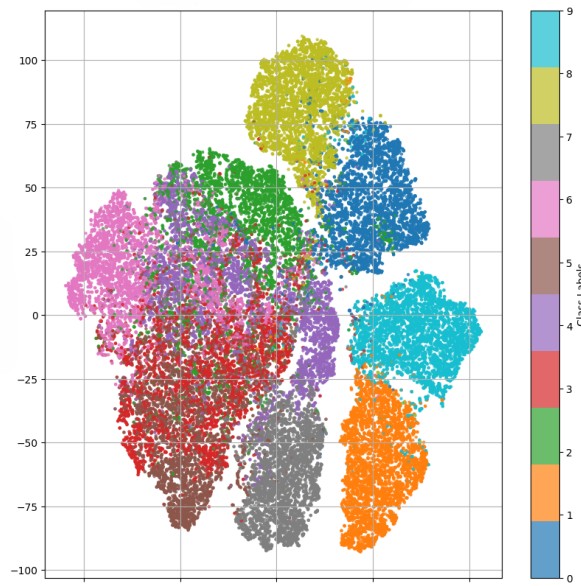

Figure 10: The t-SNE projected class distribution of CIFAR-10 within the latent space of ResNet-18 trained by PGD-AT.

