# OpenReview forum: "New Paradigm of Adversarial Training: Breaking Inherent Trade-Off between Accuracy and Robustness via Dummy Classes"
_ICLR.cc/2025/Conference — Submitted to ICLR 2025_

### Official Review · Reviewer_dTK9 · 2024-10-28

**Soundness:** 2
**Presentation:** 3
**Contribution:** 2
**Rating:** 5
**Confidence:** 4

**Summary:**

This paper proposes a new adversarial training method (DUmmy Classes-based Adversarial Training, DUCAT). DUCAT relaxes the strong assumption in adversarial training that requires clean samples and adversarial samples to be classified into the same category via introducing dummy classes. DUCAT can improve the model's adversarial robustness while achieving higher clean accuracy than existing AT methods such as PGD, TRADS, MART, and Consistency-AT.

**Strengths:**

This paper reveals that always-failed samples widely exist in existing AT methods and points out that this is caused by the overly strong assumption of AT on clean samples and adversarial samples. The corresponding dummy class method can improve the robustness of the model and, compared with existing adversarial training methods, shows an improvement in clean accuracy.

The organization and writing quality of the paper are both good.

**Weaknesses:**

1. The paper's title and the main claim is somewhat misleading (Breaking Inherent Trade-Off between Accuracy and Robustness). I don't think the proposed method indeed breaks the trade-off between accuracy and robustness. As shown in Table 1, the clean accuracy obtained via DUCAT is still significantly lower than standard training (for example, ResNet-18 can easily achieve accuracy higher than 93% under standard training). I suggest authors rephrase their claim to more accurately rephrase their claim to more accurately reflect the improvement over existing adversarial training methods, and also provide the clean accuracy of standard training for reference.
2. The use of soft labels instead of hard labels to enhance adversarial training performance somewhat limits the novelty of this paper; please refer to [1, 2, 3, 4], where [1] discusses injecting more learned smoothing during adversarial training, [2] proposes generating soft labels instead of hard one-hot labels to achieve more robust models, and so forth. Additionally, I believe that the one-versus-the-rest loss proposed in [4] already shares a similarity with the ideas presented in this paper.


[1] Robust Overfitting may be mitigated by properly learned smoothening (ICLR2021)

[2] Balance, Imbalance, and Rebalance: Understanding Robust Overfitting from a Minimax Game Perspective (NeurIPS2023)

[3] Annealing Self-Distillation Rectification Improves Adversarial Training (ICLR2024)

[4] Switching One-Versus-the-Rest Loss to Increase the Margin of Logits for Adversarial Robustness (preprint)

**Questions:**

I believe the authors should reconsider the terms in which they claim the contributions of the article. Instead of stating that it has broken the inherent trade-off between accuracy and robustness, this paper should more accurately be described as reducing the loss of clean accuracy while enhancing adversarial robustness.

Also, I would like the authors to elaborate on the differences and major innovations in their approach and related work (please refer to the weaknesses session 2).

---

> ### Author Response · Authors · 2024-11-25
> **Response to Reviewer dTK9 (1/3)**
>
> Thank you for your comments. We are glad that you find our manuscript is well organized and written. Below we respond to each point of your concerns.
>
> ---
>
> **W1&Q1: The clean accuracy obtained via DUCAT is still lower than standard training, so I don't think it breaks the trade-off between accuracy and robustness. I suggest authors rephrase their terms to reflect the improvement and claim their contributions. It should more accurately be described as reducing the loss of clean accuracy while enhancing adversarial robustness. I also suggest authors provide clean accuracy of standard training for reference.**
>
> Thank you for carefully pointing out this. We understand what you are concerned about. However, we respectfully think this is just an unfortunate difference in interpreting what is "breaking the inherent/current trade-off", which is a _**presentation problem that could be easily refined**_ and _**would not impact the contributions**_ of this work in essence.
>
> Specifically, we understand you believe that, beginning from a clean accuracy achieved by standard training (say, 93% for ResNet-18 on CIFAR-10), only if an AT method can improve robust accuracy without significantly reducing such a clean accuracy, we can say it "breaks the trade-off". Of course, this is also reasonable. But please kindly consider that, beginning from the clean and robust accuracies (e.g., as in Table 1, for PGD-AT, clean accuracy=82.92%, robust accuracy=46.74%) that already achieved by existing methods (which demonstrate the current trade-off), now that DUCAT can improve both of them at the same time (e.g., respectively 82.92%->88.81% and 46.74%->58.61%), instead of increasing one while decreasing another, couldn't this be seen as another kind of "breaking the inherent/current trade-off"?
>
> That's why as you may have already kindly noticed, when we use "break", it is usually followed by "the inherent trade-off" or "the current trade-off" (with only one exception among the whole manuscript), because they specifically refer to the undesirable current trade-off results achieved by existing methods. Also as in Figure 5, DUCAT is at a clearly different trade-off level compared with SOTA related works in the past five years (you may kindly find this by sampling different diagonal lines drawing from top left to bottom right). So all in all, when preparing this manuscript, we decided to try to claim that DUCAT can break the current (or original/old) trade-off, while it still suffers from a new trade-off (but at a much better level than before). We had also discussed this in _**Appendix D. Limitations (lines 1012-1017)**_.
>
> Still, the above response is just to explain the reason why we wrote like this before, instead of fighting for it. We agree that compared with existing AT methods, DUCAT "reduces the loss of clean accuracy while enhancing adversarial robustness" as you kindly summarized. So we sincerely believe there should be no significant disagreement between you and us regarding the essential contribution of this work, and what we are talking about is just something about the presentation. Now that you have raised this question, we are surely willing to refine it to eliminate underlying confusion. Fortunately, this can be finished by _**simply replacing all the "break" w.r.t. the trade-off to "release"**_ in our manuscript. Such a slight refinement would _**not impact the validity of any statements, arguments or contributions**_ we have made in this work, so we respectfully do not think this question should anyhow be a serious objection.
>
> We sincerely solicit your opinion of this revision plan (also including providing the clean accuracy of standard training for reference as you kindly suggested). If this would be fine for you, we would immediately make this revision. While if you have any further concerns, we would also be glad to respond.

---

> ### Author Response · Authors · 2024-11-25
> **Response to Reviewer dTK9 (2/3)**
>
> **W2&Q2: The use of soft labels instead of hard labels to enhance AT somewhat limits the novelty of this paper; please refer to [1, 2, 3, 4], where [1] discusses injecting more learned smoothing during AT, [2] proposes generating soft labels instead of hard one-hot labels to achieve more robust models, and so forth. Additionally, I believe the one-versus-the-rest loss [4] shares a similarity with the ideas presented in this paper. I would like the authors to elaborate on the differences and major innovations in their approach and the above related works.**
>
> Thank you for recommending these previous works. However, we respectfully think _**none**_ of them involve the _**main innovations of our work**_ or contribute _**experimental results that are competitive**_ to ours, so these previous works should also not oppose the novelty and contributions of our work.
>
> First of all, the most important innovation of our work is to suggest that, the current AT paradigm, which assumes to learn the corresponding benign and adversarial samples as the same class, inappropriately combines clean and robust objectives that can be essentially inconsistent. This is a _**paradigm (or task) definition-level innovation**_ that has _**never**_ been studied before (certainly also not in [1, 2, 3, 4]).
>
> Secondly, it seems only [3] among [1, 2, 3, 4] mentions soft labels. Still, please kindly note that our particular soft label is _**similar in neither motivation nor specific implementation**_ to the ones in previous works including [3]. Specifically, our novel _**motivation**_ is "to explicitly bridge corresponding original and dummy classes as the suboptimal alternative target of each other, so that their separation also becomes learnable _**(lines 138-140)**_", instead of the conventional "promoting learning from the soft target, as a better guidance mechanism that reflects the underlying distribution of data [3]". Then for _**specific implementation**_, different from conventional soft labels combining one-hot target and crafted distribution just as [3], our particular soft labels combine two one-hot targets namely as the primary target and the alternative target **_(lines 348-354)_**, which is also proposed for the first time. We greatly appreciate your question reminding us to elaborate on the particularity of our soft labels. We plan to _**refer to our soft label uniformly as a new name "two-hot soft label"**_ in our revised version to highlight its innovation.
>
> Finally, the _**experimental results**_ of [1, 2, 3, 4] are simply _**not competitive**_ to ours. For your convenience, below we showcase a comparison on CIFAR-10 and ResNet-18, where the great advantages of our results also imply the essential difference between DUCAT and previous works. Please kindly note that [3] and [4] were respectively cited as "Wu et al., 2024" and "Kanai et al., 2023" in our manuscript, both of them already served as one of the 14 SOTAs we compared with in _**Section 3.2**_ and _**Appendix A.3**_, so we just copy the results from there. On the contrary, [1, 2] are new to our manuscript, so we take the results from their original paper (where [2] even uses the more powerful PreActResNet-18, but still achieves weaker results than DUCAT on ResNet-18).
>
> |   Method    |  Clean Acc  | Robust Acc |
> | --------------- | --------------- | --------------- |
> |        [1]        |      84.65     |     52.14      |
> |        [2]        |      78.71     |     51.49      |
> |        [3]        |      82.41     |     50.38      |
> |        [4]        |      81.90     |     49.40      |
> |  **Ours**  |  **88.81**  | **58.61**  |

---

> ### Author Response · Authors · 2024-11-25
> **Response to Reviewer dTK9 (3/3)**
>
> **W2&Q2 (Cont.)**
>
> Apart from the most important three points of response above, here are also some minor points w.r.t. [1, 2, 4] respectively:
>
> - [1] specifically studies model smoothing technology, which is not a perspective we studied in our work (actually we did ever not use the word "smooth" when talking about DUCAT). The principle of DUCAT's effectiveness is not similar to "injecting more learned smoothing during AT [1]", because it does not even "learn" smoothing. Instead, it is our more reasonable learning paradigm (or task setting) that from the beginning prevents the model from struggling to learn hard patterns. Of course, if being asked whether the model becomes smoother or not as a result, the answer would be yes. However, this should not mean that model smoothing limits our novelty, otherwise it would also limit the novelty of most subsequent works.
>
> - Just as its title implies, [1] aims at robust overfitting, which is a relevant but somewhat different topic with the accuracy-robustness trade-off. Although [1] indeed showcases some concurrent improvements in accuracy and robustness (still, weaker than ours), it mainly studies the generalization gap between training robust accuracy and test robust accuracy, especially regarding its sharp degradation after certain epochs of AT (e.g., after learning rate decay). In contrast, in our work, we study the trade-off problem from the task-level inconsistency between clean learning and robust optimization.
>
> - We could not find any content about soft labels or relevant concepts in [2]. Please kindly correct us if this is our issue.
>
> - After the above discussion about conventional soft labels and our "two-hot soft labels", we believe you are professional and perceptive enough to find now that [4] actually shares a similarity more with the ideas of conventional soft labels rather than with ours. Because [4] studies the margins between logits for the true label and the other labels, which is somewhat relevant to soft labels combining one-hot target (considering true label) and crafted distribution (considering other labels). On the other hand, again, the huge difference in experimental results also does not support much possible similarity between DUCAT and [4].

---

> ### Comment · Reviewer_dTK9 · 2024-11-27
> **Response to Authors**
>
> Thank you for your response. While I still hold my original view on the use of terms like "inherent trade-off" and "soft label," I would reconsider my score if the authors revised the terminology as suggested in their response. Regardless, it is indeed a significant contribution to see a substantial improvement in model robustness, which has been rare in the field of adversarial robustness for quite some time.

---

> > ### Author Response · Authors · 2024-11-28
> > **Thank you for the kind feedback and reconsideration**
> >
> > Thank you very much for your detailed feedback. We are greatly encouraged by your favorable comment valuing the substantial robustness improvement of DUCAT, and we appreciate your kind reconsideration of the score. We understand and respect your view on the use of specific terms, and have revised our manuscript based on your kind suggestions, as the items 1, 2, 4 and 8 in our General Response above. We would be glad to hear from you if there are any points that remain to be further improved.

---

### Official Review · Reviewer_QPXH · 2024-11-03

**Soundness:** 3
**Presentation:** 3
**Contribution:** 3
**Rating:** 6
**Confidence:** 4

**Summary:**

This paper proposed DUCAT, a new adversarial training paradigm that leverages dummy classes to achieve a better trade-off between adversarial robustness and clean accuracy. The introduced dummy classes relax the overstrict label assignment. As a plug-and-play method, DUCAT can be applied to existing adversarial training pipelines to help further improve performance. The effectiveness of DUCAT is demonstrated by extensive experiments across different datasets.

**Strengths:**

- The intuition behind introducing dummy classes is making sense to me.
- The paper is well-written and easy to follow.
- The proposed methods are evaluated on different datasets, baseline methods and network architectures.

**Weaknesses:**

- I agree with the design of DUCAT and its effectiveness when applied to existing adversarial training methods, but a non-negligible fact is that the performance is still far away from methods like [1, 2, 3], which obtain higher clean and robust accuracy. Although DUCAT is a good plug-and-play method for adversarial training, I think the value is still limited if it cannot bring those AT-based baseline methods to a closer level to more advanced methods.
- Some notations are not clearly explained. Please see the questions below.

[1] Wang, Z., Pang, T., Du, C., Lin, M., Liu, W., & Yan, S. (2023, July). Better diffusion models further improve adversarial training. In International Conference on Machine Learning (pp. 36246-36263). PMLR.

[2] Rebuffi, S. A., Gowal, S., Calian, D. A., Stimberg, F., Wiles, O., & Mann, T. (2021). Fixing data augmentation to improve adversarial robustness. arXiv preprint arXiv:2103.01946.

[3] Sehwag, V., Mahloujifar, S., Handina, T., Dai, S., Xiang, C., Chiang, M., & Mittal, P. (2022). ROBUST LEARNING MEETS GENERATIVE MODELS: CAN PROXY DISTRIBUTIONS IMPROVE ADVERSARIAL ROBUSTNESS?. In 10th International Conference on Learning Representations, ICLR 2022.

**Questions:**

- What is $\mathcal{P}(x_i)$ in Eq. 2? The distribution of adversarial examples of $x_i$?
- What are $\dot{\mathbf{y}}$ and $\ddot{\mathbf{y}}$ in Eq. 4? Do they stand for probabilities of $C$ real classes and $C$ dummy classes respectively? Are you actually doing a decomposition $\mathbf{y} = \dot{\mathbf{y}} + \ddot{\mathbf{y}}$?
- What is $h_\theta^{Dummy}(\cdot)$? Another neural network? If I understand Fig. 1 correctly, the output representation of the second last layer is unchanged, and you just enlarge the dimensionality of the last layer? Can you clarify if you are using another neural network or just enlarging the last layer?
- How do you set the dummy classes for each example?
- I am confused about the word "SOTAs" used in Sec 3.2. If it means those aiming at releasing the trade-off between accuracy and robustness, then we can find a bunch of methods overperforming them on RobustBench[4]. If it specifically refers to AT-based methods, then like what I mentioned in the weaknesses above, the value of DUCAT is limited as we can find more advanced methods. In fact, methods like [1] also involve adversarial training. If DUCAT is proved able to improve the performance of some of those methods, its value will be underscored.

[4] https://robustbench.github.io/

---

> ### Author Response · Authors · 2024-11-25
> **Response to Reviewer QPXH (1/3)**
>
> Thank you for your detailed reviews and also for valuing the design of DUCAT. We hope the following response could address your questions.
>
> ---
>
> **W1&Q5: I am confused about the word "SOTAs" in Sec 3.2. If it means those aiming at releasing the trade-off between accuracy and robustness, then we can find a bunch of methods overperforming them on RobustBench[4]. If it specifically refers to AT-based methods, then the value of DUCAT is still limited if it cannot bring AT-based baselines to a closer level to more advanced methods like [1, 2, 3], which obtain higher clean and robust accuracy. In fact, methods like [1] also involve AT. If DUCAT is proven able to improve the performance of some of those methods, its value will be underscored.**
>
> Thank you for caring about the value of DUCAT in the big picture and recommending the related works. However, we respectfully think we do _**involve all the SOTAs comparable to DUCAT under fair settings**_, considering all those aiming at releasing the trade-off between accuracy and robustness (including the ones in RobustBench [4]). We did not include works with synthetic data like [1, 2, 3] in our original manuscript because we think they focus more on _**data**_ rather than _**algorithms**_, and there are certain concerns about _**fairness**_, especially compared with conventional AT. Still, if this is not a problem, then we could also _**easily outperform them**_ by simply utilizing the open-source synthetic data from [1], just as the _**additional experimental results**_ we showcase. Below we would like to detail each of these points in order.
>
> First of all, we respectfully point out that, you could _**not**_ actually "find a bunch of methods overperforming DUCAT on RobustBench [4]". Please kindly note the specific threat model and experimental settings for a fair comparison, especially regarding the _**model structure**_ and the _**usage of extra/synthetic data**_. For instance, on CIFAR-10 and ResNet-18 without extra/synthetic data, the current SOTA on RobustBench [4] is DAJAT [R1], which had already been included in our 14 SOTAs. To the best of our knowledge, for the other specific structures/datasets in our work, there should also be no more SOTAs left. Please kindly correct us if there are indeed any other SOTAs we missed under such settings.
>
> Then we would like to explain why we sincerely think it is reasonable not to worry about the related works with extra/synthetic data. In short, they first simply violate the _**default fairness assumption**_ of AT. More importantly, they are actually _**not in competition**_ with conventional AT methods. Instead, they could be easily integrated into them.
>
> [R1] Addepalli et al. Efficient and Effective Augmentation Strategy for Adversarial Training. In NeurIPS, 2022.

---

> ### Author Response · Authors · 2024-11-25
> **Response to Reviewer QPXH (2/3)**
>
> **W1&Q5 (Cont.)**
>
> - For fairness, firstly, it is widely recognized that _**extra data should not be included**_ when comparing AT techniques [R2], as it could basically improve all AT methods to the same degree. So it is not about specific AT algorithms but data. For _**synthetic data**_, although they are fairly based on the same training dataset, the problem is the _**dramatic additional computational cost**_ they rely on. For instance, they need to _**train a diffusion model first**_ [1, 2, 3], which is even much more time-consuming than training the robust model itself (e.g., training a ResNet-18 with PGD-AT on CIFAR-10 typically needs only about 4 hours with RTX4090, while training a DDPM (a common types of diffusion) model on CIFAR-10 usually needs up to 30 hours with A100). Although it is argued that both training generative models and sampling from them is a one-time cost [3], that is only true for experimental scenarios with standard datasets, but certainly _**not for specific real-world practices**_. Also, synthetic data-based methods rely on _**extra standard-trained models for pseudo-labeling**_ of the generated unlabeled data [1, 2, 3]. Besides, their training epochs and batch size are significantly larger than typical AT [1, 2, 3], which respectively means the needs for _**more training time and the RAM resource**_. We sincerely believe that, for fairness of comparison, we should always _**take these costs into consideration**_ when discussing the impressive results achieved by synthetic data-based methods.
>
> - Sometimes, a more important philosophy behind a fairness assumption is that, methods following the assumption would be _**not directly in competition**_ with the ones not, and here is such a case. While AT methods focus more on the advancement in _**algorithms**_, the synthetic data-based methods contribute more to the _**data**_ aspect. As a result, these two kinds of works would not degrade the contributions of each other, but could be integrated. Actually, the recommended [1, 2, 3] _**all involve AT**_. More specifically, all of them are essentially _**TRADES (i.e., one of our baselines) + different data augmentation approaches with diffusion models**_. So although DUCAT, just as you said, could not bring AT to a closer level to [1, 2, 3] _**just by itself**_, there is simply no need to do that. Given that DUCAT was proven effective in improving the AT baselines, naturally it could also improve synthetic data-based methods _**built upon such baselines**_ (again, like [1, 2, 3] upon TRADES).
>
> To support our discussion above, we are also glad to provide preliminary additional experimental results, demonstrating that by simply using the open-source 1M CIFAR-10 synthetic data from [1] without other tricks it adopts (e.g., weight averaging and cyclic learning rate schedule) and the more training epochs it needs (i.e., at least 400 epochs), we can easily train a WideResNet-28-10 achieving _**better clean accuracy and competitive robust performance**_ compared with [1], by our DUCAT with just the same training details in our manuscript (due to the limited time, as [1] is the SOTA among [1, 2, 3] and relevant works, here we only compare with it and, also, with its lightest settings).
>
> | Method | Clean | Auto-Attack |
> |-------------|---------|------------------|
> |      [1]     | 91.12 |      63.35       |
> |    Ours    | 92.60 |     62.74       |
>
> Considering the _**simplicity and computational-friendliness**_ of DUCAT, we sincerely believe even this result itself is considerable. But please kindly note that, similarly for the time reason, we have not tried a more promising direction, which is to directly integrate DUCAT to [1] with its training settings. We sincerely promise to provide further results in our future revision if we are given a chance, including directly integrating DUCAT to [1], trying other settings of [1] (e.g., with more synthetic data), and additionally involving [2, 3].
>
> [R2] Pang et al. Bag of Tricks for Adversarial Training. In ICLR, 2021.

---

> ### Author Response · Authors · 2024-11-25
> **Response to Reviewer QPXH (3/3)**
>
> **Q1: What is** $\mathcal{P}(\mathbf{x}\_{i})$ **in Eq. 2? The distribution of adversarial examples of $\mathbf{x}\_{i}$?**
>
> Basically yes. More precisely, $\mathcal{P}(\mathbf{x})$ is a pre-defined perturbation set, as described in _**lines 235-236**_. This is a standard description that is consistent with the previous work  PGD-AT, which means $\mathbf{x}\_{i}' \in$ $\mathcal{P}(\mathbf{x}\_{i})$ could be any possible adversarial sample of $\mathbf{x}\_{i}$ w.r.t. specific adversary represented by $\mathcal{P}(\mathbf{x})$.
>
> ---
>
> **Q2: What are** $\dot{\mathbf{y}}\_{i}$ **and** $\ddot{\mathbf{y}}\_{i}$ **in Eq. 4? Do they stand for probabilities of** $C$ **real classes and** $C$ **dummy classes respectively? Are you actually doing a decomposition** $\mathbf{y} = \dot{\mathbf{y}} + \ddot{\mathbf{y}}$**?**
>
> Yes, we are doing a decomposition $\mathbf{y} = \dot{\mathbf{y}} + \ddot{\mathbf{y}}$, just as suggested in _**line 265**_. Still, please kindly note that $\dot{\mathbf{y}}$ and $\ddot{\mathbf{y}}$ are standing for one-hot target rather than probabilities, while with the value of the hot element less than 1. For example, given a three-class classification task (i.e., $C=3$), we may have a case that $\mathbf{y}\_{i}=[0, 1, 0]$, $\dot{\mathbf{y}}\_{i}=[0, 0.9, 0]$ and $\ddot{\mathbf{y}}\_{i}=[0, 0.1, 0]$. One reason for the specific formulation of Eq. 4 is to highlight that only a pair of elements in the same place of $\dot{\mathbf{y}}$ and $\ddot{\mathbf{y}}$ can be correspondingly valued, while all of the other elements remain 0.
>
> ---
>
> **Q3: What is $h_{\theta}^{\text{Dummy}}(\cdot)$? Another neural network? If I understand Fig. 1 correctly, the output representation of the second last layer is unchanged, and you just enlarge the dimensionality of the last layer? Can you clarify if you are using another neural network or just enlarging the last layer?**
>
> We are glad to see that you precisely understand Fig. 1. We just enlarge the dimensionality of the last layer. So $h_{\theta}^{\text{Dummy}}(\cdot)$ is not to refer to another neural network, but to represent the output logits of the enlarged model w.r.t. the $C$ additional dummy classes. This is to be consistent in notations with DuRM, which is the only previous work involving dummy classes (but focusing on the standard generalization of ERM).
>
> ---
>
> **Q4: How do you set the dummy classes for each example?**
>
> Theoretically, the specific one-to-one correspondences between real and dummy classes are customizable. But for simplicity, in our experiments, we uniformly assign the first $C$ output dimensions for real classes (from real class 0 to real class $C-1$) and the other $C$ dimensions for dummy classes in the same order. Taking CIFAR-10 as an example, that means in the enlarged 20 output dimensions, the 1st dimension is for the real class 0 and the 11th dimension is for the dummy class 0, and so on. Thus, for each benign example belonging to a specific real class, as well as the adversarial sample generated from this benign example, their specific dummy class is set to be the dummy class that corresponds to their real class.

---

> ### Comment · Reviewer_QPXH · 2024-11-26
> **Response to the Authors**
>
> I greatly appreciate the responses from the authors, which have addressed many of my concerns. Besides, I agree that it is fairer to compare different robust training methods under similar levels of computational cost, but it still falls behind those methods using more computational resources. What I mean by "the value is still limited if it cannot bring those AT-based baseline methods to a closer level to more advanced methods" is that if high robustness is a non-negligible factor, we still need to choose those methods. However, I do recognize the contribution of DUCAT in addition to adversarial training. Therefore, I decided to raise my rating to 6.

---

> > ### Author Response · Authors · 2024-11-27
> > **Thank you for raising the rating**
> >
> > Thank you very much for raising the rating. We sincerely appreciate your timely feedback and further clarification. We are glad to see your agreement with the contribution of our DUCAT, and we are willing to incorporate your kind suggestion as a reference for readers :)

---

### Official Review · Reviewer_r1aS · 2024-11-03

**Soundness:** 2
**Presentation:** 3
**Contribution:** 2
**Rating:** 6
**Confidence:** 4

**Summary:**

In this paper, the authors address the ongoing trade-off issue in adversarial training-based methods. To mitigate this problem, they introduce a dummy class and soft labeling techniques into the existing adversarial training framework. A novel adversarial training loss function is designed to improve the model’s trade-off between accuracy and robustness. The proposed method is evaluated on three datasets and compared against four state-of-the-art adversarial training approaches. Additionally, the authors analyze the impact of various hyperparameters on model performance.

**Strengths:**

1. Addresses a Key Limitation in Adversarial Training: The paper introduces a new adversarial training paradigm, DUCAT, which aims to break the inherent trade-off between clean accuracy and robustness, a well-known limitation in existing methods

2. Plug-and-Play Applicability: DUCAT is presented as a flexible method that can be integrated with existing adversarial training frameworks, potentially making it easier to adopt in practical applications without major redesigns.

**Weaknesses:**

1. Lack of In-depth Analysis of Method Efficacy: The proposed DUCAT method primarily leverages the concepts of dummy classes and label smoothing. However, previous studies (e.g., [1], [2], [3], [4]) have explored the effectiveness of either dummy classes or label smoothing to enhance robustness or accuracy. A more thorough analysis is needed to clarify the unique contribution of DUCAT in comparison to these established works.

   [1] Label Smoothing and Logit Squeezing: A Replacement for Adversarial Training? https://arxiv.org/abs/1910.11585

   [2] Frustratingly Easy Model Generalization by Dummy Risk Minimization https://arxiv.org/pdf/2308.02287

   [3] When Does Label Smoothing Help?

    [4] MIXUP INFERENCE: Better Exploiting Mixup to Defend Adversarial Attacks https://openreview.net/pdf?id=ByxtC2VtPB

2. Limited Empirical Evidence: The experimental results, as seen in Table I, show only marginal improvements of the proposed DUCAT method over TRADES. Further analysis is recommended to understand the limited performance gains.

**Questions:**

1. How does the proposed DUCAT method compare in depth to the above-mentioned related works? Could you elaborate on the unique aspects of its effectiveness?

2. What might be the reasons behind the limited performance improvement of DUCAT over TRADES?

3. Could you provide a performance comparison between the proposed DUCAT method and the technique outlined in [4]?

     [4] MIXUP INFERENCE: BETTER EXPLOITING MIXUP TO DEFEND ADVERSARIAL ATTACKS https://openreview.net/pdf?id=ByxtC2VtPB

---

> ### Author Response · Authors · 2024-11-25
> **Response to Reviewer r1aS (1/2)**
>
> Thank you for your comments and questions. We are happy to see your recognition of our two strengths. We hope you would find our response useful in addressing your questions.
>
> ---
>
> **W1&Q1: DUCAT leverages dummy classes and label smoothing. However, previous studies have explored the effectiveness of either dummy classes [2] or label smoothing [1], [3], [4] to enhance robustness or accuracy. How does DUCAT compare in depth to these established works? Like unique aspects of its effectiveness or unique contributions?**
>
> Firstly, for the dummy class, [2] is the only previous study and we had compared DUCAT with [2] in detail in _**Section 2.2.2 (lines 291-306)**_. In short, [2] aims at facilitating standard generalization of ERM, while DUCAT aims to release the tension between clean and robust objectives in AT; [2] typically adds only 1 or 2 dummy classes, but DUCAT always adopts the same number of dummy classes as original classes and explicitly builds one-to-one correspondences between them; [2] just contributes trivial improvement to AT, yet DUCAT achieves SOTA performance.
>
> Secondly, we respectfully think _**DUCAT does not leverage label smoothing**_. Because _**the particular soft label adopted by DUCAT is completely different**_ from the conventional soft labels used for label smoothing as [1], [3]. Previously, "label smoothing converts one-hot label vectors into one-warm vectors that represents a low-confidence classification [1]". But for DUCAT, our particular soft label is neither built in a "one-warm" format nor aiming at "a low-confidence classification". Specifically, different from conventional soft labels combining one-hot target and uniform/crafted distribution [1], [3], [R1], [R2], DUCAT builds particular soft labels that combine two one-hot targets namely as the primary target and the alternative target _**(lines 348-354)**_. This is to explicitly bridge corresponding original and dummy classes as the suboptimal alternative target of each other, so that their separation also becomes learnable _**(lines 138-140)**_", rather than to promote learning from the soft target as previous works including [1], [3]. This unique principle is proposed for the first time, strongly supporting the outstanding effectiveness of DUCAT. We greatly appreciate your question reminding us to highlight the particularity of our soft labels as a novel contribution. We plan to realize this by _**referring to our soft label uniformly as a new name "two-hot soft label"**_ in our revised version.
>
> Finally, as an _**inference time**_ work, [4] seems to impact little on the unique aspects of DUCAT's effectiveness or contributions, because DUCAT is a training-based method. We defer the detailed discussion to the Q3 below.
>
> [R1] Szegedy et al. Rethinking the inception architecture for computer vision. In CVPR, 2016. https://www.cv-foundation.org/openaccess/content_cvpr_2016/html/Szegedy_Rethinking_the_Inception_CVPR_2016_paper.html
>
> [R2] Wu et al. Annealing self-distillation rectification improves adversarial training. In ICLR, 2024. https://openreview.net/forum?id=eT6oLkm1cm

---

> ### Author Response · Authors · 2024-11-25
> **Response to Reviewer r1aS (2/2)**
>
> **W2&Q2: The experimental results in Table 1 show only marginal performance improvement of DUCAT over TRADES. What might be the reasons behind this limited empirical evidence? Further analysis is recommended to understand it.**
>
> First, we would like to respectfully point out that, precisely speaking, the experimental results show limited improvement of _**TRADES+DUCAT over TRADES**_, rather than "DUCAT over TRADES". In other words, they just reflect the limitation of DUCAT as a _**plug**_, instead of DUCAT as an _**independent AT method**_. Actually, DUCAT itself as an AT method (i.e., PGD-AT+DUCAT in Table 1) significantly outperforms TRADES. We more detailed discussed this when responding to W1 from Reviewer 9r4X, please kindly have a look if you are interested :)
>
> Then back to your question. It could now be described as: _**Why DUCAT helps little in enhancing TRADES?**_ Well, we believe it is because of the _**particular training adversarial samples used by TRADES**_. Specifically, different from most AT methods including the other three benchmarks (PGD-AT, MART and Consistency-AT) that by default generate training adversarial samples by PGD-10, TRADES particularly crafts training adversarial samples through maximizing its own KL-divergence regularization term [R3]. Because TRADES defines a "boundary error" measured through this KL term, which occurs when specific data points are sufficiently close to the decision boundary that can easily cross it under slight perturbation [R3], so it makes sense to accordingly craft training adversarial samples to reduce this error. But as a consequence, these training adversarial samples naturally focus more on boundary establishment instead of reasonable data distribution patterns, which predictably degrades our DUCAT, as DUCAT assumes most of the adversarial samples from one class should belong to the corresponding dummy class.
>
> We had mentioned this in _**Appendix D. Limitations (lines 1018-1022)**_. We greatly appreciate your question reminding us about its importance. We plan to further add the above discussion in our revised version.
>
> [R3] Zhang et al. Theoretically principled trade-off between robustness and accuracy. In ICML, 2019. http://proceedings.mlr.press/v97/zhang19p
>
> ---
>
> **Q3: Could you provide a performance comparison between DUCAT and the technique outlined in [4]?**
>
> Yes, of course. We notice that [4] develops mixup inference (MI) to mixup input with random clean samples at inference time, thus shrinking and transferring the equivalent perturbation if the input is adversarial. It is an interesting and novel work that also considers outside the conventional perspective, just as ours. So we are glad to provide an additional subsection in Appendix C. Supplementary Experiments later in our revised version to involve a discussion with MI and the following performance comparison results. Thank you for your kind recommendation.
>
> Still, the performance comparison between DUCAT and MI on CIFAR-10 further showcases _**our advantage**_. The performance of MI is taken from [4] and that of DUCAT is from our Table 1 (PGD-AT+DUCAT). As the robust performance of MI is not evaluated by Auto-Attack there, we compare robust accuracy under PGD-10 adversary below, as both our work and [4] originally involve it. Please kindly note that even if the results of DUCAT here are acquired from ResNet-18 while those of MI are from more powerful ResNet-50, our DUCAT still outperforms MI.
>
> |   Method    |  Clean Acc  | Robust Acc |
> | --------------- | --------------- | --------------- |
> |      MI [4]    |      84.20     |     64.50      |
> |  **Ours**  |  **88.81** | **65.10**  |

---

> > ### Comment · Reviewer_r1aS · 2024-11-27
> > **Response to authors**
> >
> > Thank you for your comprehensive response. Your detailed explanations have effectively addressed my concerns regarding the unique contributions of DUCAT compared to existing methods, the observed performance improvements, and the comparative analysis with Mixup Inference. I appreciate the clarity and depth of your clarifications. Consequently, if the authors would revise and improve the paper accordingly, I am inclined to raise my evaluation score to 6.

---

> > > ### Author Response · Authors · 2024-11-28
> > > **Thank you for the kind feedback and inclination**
> > >
> > > Thank you very much for your kind feedback. We are happy to see your concerns are addressed and we appreciate your inclination to raise the score. We have revised our manuscript according to your kind suggestions, as the items 2, 6 and 7 in our General Response above. We would be glad to make further improvements if there is still something not up to your requirements.

---

### Official Review · Reviewer_9r4X · 2024-11-03

**Soundness:** 3
**Presentation:** 3
**Contribution:** 3
**Rating:** 6
**Confidence:** 2

**Summary:**

Pointing out that the existing AT methods enforce the model to learn benign and adversarial samples are belong to the same class lead to the trade-off between clean and adversarial accuracy, this paper introduced a new AT paradigm introducing additional dummy classes for certain adversarial samples that differ in distribution from the original ones to relax the assumption of existing AT methods. Rather than strictly separating the class for benign and adversarial examples, this paper construct soft labels to explicitly bridge corresponding original and dummy classes as the suboptimal alternative target of each other, leading the model to learn separation between them. Experimental results across different baselines and tasks show effectiveness of the proposed method.

**Strengths:**

- This paper has strong motivation of tackling trade-off between clean and robust accuracy of existing AT methods, demonstrating existing methods all fail to some adversaries that proves the common deficiency of existing works' learning objectives.
- This paper is well written and easy to follow.

**Weaknesses:**

- The proposed method seem to be incremental compared to TRADES.
   - Although the proposed method shows impressive improvement on PGD-AT, MART, and Consistency-AT, compared to the most important baseline that addressed the trade-off issue for the first time, TRADES, DUCAT sacrificed clean accuracy more than the improved robustness as shown in Table 1, especially on CIFAR-10. Also, the improvements achieved on CIFAR-100 and Tiny-ImageNet from the proposed method (DUCAT) are marginal while it requires additional computational resources compared to TRADES as shown in Figure 6. Based on this observation, it would be better to provide more supporting arguments on DUCAT compared to TRADES.
- Since this paper insists that if the adversarial examples generated from one model prove to be effective, they are might also effective and transferred to the different models. In responses, it would be beneficial to show the adversarial and standard accuracy under the transfer attack scenario to further strengthen the proposed method. The experiments can be done in the Figure 3 showing transfer attack scenarios across different training algorithms, where adversarial examples generated from ResNet18 model trained on DUCAT can be transferred to baseline methods, or vice versa. Rather than bar plot, it would be clearer to visualize the performance on CIFAR-10 and CIFAR-100 in the format of table.

**Questions:**

- In figure 2, there is different tendency in ratio between all failure/all worked cases for different class, why different classes show different tendency? It would be helpful to provide a more detailed analysis of the class-specific differences along with the dataset name and explain which factor of each class leads to different distributions for the results. Please examine properties of the different classes, for example, they could visual complexity, number of entities within example, what are the main differences captured by models between adversarial examples and clean examples using PCA analysis, which might affect to the tendencies in failure/success rates.

---

> ### Author Response · Authors · 2024-11-25
> **Response to Reviewer 9r4X (1/2)**
>
> Thank you for your time and efforts in reviewing our manuscript, and also for valuing our strong motivation. Please kindly let us know if we do not address your concerns in the following response.
>
> ---
>
> **W1: The proposed DUCAT seems to be incremental compared to TRADES, the most important baseline that addressed the trade-off issue for the first time, with marginal improvements but additional computational resources. So it would be better to provide more supporting arguments on DUCAT compared to TRADES.**
>
> First, we would like to respectfully point out a misunderstanding in this comment. All the experimental results you referred to here could only reflect that _**TRADES+DUCAT**_ is incremental to TRADES, instead of "_**DUCAT**_ is incremental to TRADES". The key point is that, DUCAT as a _**plug**_ having a limitation on enhancing TRADES does not mean that DUCAT as an _**independent AT method**_ is incremental to TRADES. Given TRADES as "the most important baseline of the trade-off issue", the fair comparison should certainly focus more on whether the DUCAT method outperforms TRADES, rather than whether the DUCAT plug significantly enhances TRADES.
>
> For DUCAT as an independent AT method, we sincerely ask you to refer to the results of "_**PGD-AT+DUCAT**_" in our experiments (Table 1 & Table 2). This is quite natural because PGD-AT is the basis of most AT methods such that "PGD-AT+something" basically means "something itself" in the field of AT. Taking other benchmarks for instance, MART can be viewed as "PGD-AT+MART", and that is also true for Consistency-AT, since they are all built upon PGD-AT. Now we believe you could notice that the DUCAT method actually outperforms TRADES in both clean and robust performance on all the experimental datasets you were concerned about in the original comment. So _**DUCAT greatly beats TRADES, instead of being incremental to it**_. Also, under our impressive results, we respectfully believe that the additional training time cost of about only 8% (Figure 6) is completely acceptable.
>
> Then we would also like to additionally explain _**why DUCAT helps little in enhancing TRADES**_, which is indeed a limitation when using DUCAT as a _**plug**_. In short, it is because the _**particular training adversarial samples used by TRADES**_ may be less appropriate for building dummy clusters than standard ones (BTW, such particular training adversarial samples are also what make TRADES a minority case not built upon PGD-AT among existing AT methods). We talked about such particularity and its impact on TRADES+DUCAT when responding to Reviewer r1aS, please kindly refer to W2&Q2 there if you would like to see more details :)
>
> Besides, we had mentioned this limitation in _**Appendix D. (lines 1018-1022)**_. Greatly thanks for your kind question that reminds us about the necessity of a clearer explanation. We plan to involve the discussion above in our revised version.
>
> ---
>
> **Q1: In Fig. 2, different classes show different tendencies in the ratio between all failure/all worked cases, why? A class-specific analysis would be helpful.**
>
> Thank you for carefully noting the different tendencies and the kind suggestions. In brief, this can be explained as _**the difference in the latent distribution**_ of different class clusters w.r.t. adversarial robustness. Some classes are distributed closer to each other while some are not, which makes them easier/harder to perturb under the same adversary. A commonly used technique to project such a difference to 2-D space is t-SNE [R1], through which we showcase the class distribution of CIFAR-10 within the latent space of ResNet-18 trained by PGD-AT. The result is well-aligned with Fig. 2 and confirms the explanation above. Specifically, classes 0, 1, 7, 8, 9 are better separated from others while classes 2, 3, 4, 5, 6 are relatively worse separated, which naturally results in the defense on the former ones easier than the latter ones. As we are not able to directly post the new result figure here, we update it by _**the end of our manuscript PDF (Fig. 10)**_. Please kindly refer to it, thank you.
>
> Besides, we would like to respectfully explain that, the class-level imbalance of adversarial robustness is not a point of our work. It has been well-studied in some previous works like [R2] as an independent topic, so we might not contribute much to the imbalance topic here. Instead, we expected Fig. 2 to provide an impressive intuition to readers about the universality of the high overlap of failure cases in existing AT methods, just as we described in its caption. Thank you for carefully reminding us that such a class-wise breakdown may not be that necessary, and they may hold the reader up. We are planning to simplify Fig. 2 to a non-class-wise format in the revised version of our manuscript.
>
> [R1] Van der Maaten et al. Visualizing data using t-SNE. JMLR, 2018, 9(11).
>
> [R2] Wei et al. CFA: Class-wise calibrated fair adversarial training. In CVPR, 2023.

---

> ### Author Response · Authors · 2024-11-25
> **Response to Reviewer 9r4X (2/2)**
>
> **W2: Since the paper insists if the adversarial examples generated from one model prove to be effective, they are might also effective and transferred to different models, it would be beneficial to show the adversarial and standard accuracy under the transfer attack scenario. The experiments can be done with the settings of Figure 3.**
>
> Thank you for being interested in the point behind Figure 3, and also for your very detailed suggestions on how to implement the additional experiments and show the results appropriately. We _**conducted experiments closely following your suggestions**_, with the results illustrated by the two tables below (because of the limited time here we showcase CIFAR-10 first). Specifically, the tables respectively show the attack success rate (ASR) and the specific number of successful adversarial samples, where adversarial samples are generated from models trained on the AT methods in the row headers, and transferred to attack the models trained on the AT methods in the column headers. The two values "- / -" within each cell respectively correspond to the "Originally Effective Adversarial Samples" and "Originally Ineffective Adversarial Samples" in Figure 3. We leave out standard accuracy here as it would not change along with transfer attacks, please kindly refer to Table 1 of our manuscript for it if you are interested.
>
> In short, the results among the four baselines show the same tendency as in Figure 3. And for the additional results for DUCAT, there are two points worth noting. First, when transferring adversarial samples from others to attack the DUCAT-trained model, no matter whether for originally effective or ineffective adversarial samples, there is a drop observed for both ASR and Success Num. Second, on the contrary, when transferring adversarial samples from DUCAT to attack the other four, while the Success Nums add up to smaller than the other four since the number of originally effective adversarial samples of DUCAT is much smaller, there is an increase in ASR, especially for the originally ineffective adversarial samples. In our opinion, on the one hand, DUCAT beats more adversarial samples, so on average the ones it fails to defend (i.e., the originally effective adversarial samples) are expected to be more dangerous. On the other hand, for originally ineffective adversarial samples, many of them are ineffective because of the new paradigm of DUCAT. So when they are transferred to attack the four baselines under the conventional AT paradigm, a significantly higher proportion of them can be successful.
>
> | ASR(%)         | PGD-AT      | TRADES     | MART       | Consistency-AT | DUCAT      |
> |----------------|-------------|------------|------------|----------------|------------|
> | PGD-AT         | 100/0       | 81.70/3.78 | 82.81/4.87 | 80.06/1.19     | 77.28/0.82 |
> | TRADES         | 81.37/3.18  | 100/0      | 84.09/4.45 | 75.97/2.76     | 75.47/2.36 |
> | MART           | 77.41/2.98  | 79.40/3.11 | 100/0      | 72.05/2.51     | 70.24/1.86 |
> | Consistency-AT | 86.25/2.71  | 81.82/5.46 | 82.53/6.50 | 100/0          | 79.78/1.64 |
> | DUCAT          | 88.92/12.37 | 80.24/9.75 | 83.96/9.84 | 84.23/10.18    | 100/0      |
>
> | Attack Success Num | PGD-AT   | TRADES   | MART     | Consistency-AT | DUCAT    |
> |--------------------|----------|----------|----------|----------------|----------|
> | PGD-AT             | 4869/0   | 3978/194 | 4032/250 | 3898/61        | 3763/42  |
> | TRADES             | 3861/167 | 4745/0   | 3990/234 | 3605/145       | 3581/124 |
> | MART               | 3584/160 | 3676/167 | 4630/0   | 3336/135       | 3252/100 |
> | Consistency-AT     | 4009/145 | 3803/292 | 3836/348 | 4648/0         | 3708/88  |
> | DUCAT              | 3092/807 | 2790/636 | 2919/642 | 2929/664       | 3477/0   |
>
> Besides, we would like to respectfully and warmly remind you that, while the above content is indeed a point that could be further studied behind Figure 3, the main purpose of Figure 3 is to show the wide existence of always-failed cases crossing robust models under the current AT paradigm (which is one of the important parts of the motivation of our work to propose the new paradigm), rather than to insist on any specific opinions w.r.t. transfer attacks. We would appreciate it if you could also consider this.

---

> > ### Comment · Reviewer_9r4X · 2024-11-27
> >
> > Thank you for your detailed responses and clarification. Still, I think the performance improvement compared to computational cost seems to be incremental so I will increase my score to 6.

---

> > > ### Author Response · Authors · 2024-11-28
> > > **Thank you for increasing the score**
> > >
> > > Thank you very much for increasing the score. We respect your feedback and opinion.

---

### Author Response · Authors · 2024-11-28
**General Response - Sincere Thanks and Revision Summary**

We would like to sincerely thank all the reviewers for their time and efforts in reviewing our manuscript and sharing thoughtful comments, as well as giving timely feedback during the rebuttal.

We have revised and submitted our manuscript incorporating the suggestions of the reviewers, where the modifications are highlighted in red for clarity, including:

1) Replacing all the "break" w.r.t. the trade-off to "release" (Reviewer dTK9)

2) Referring to our soft label uniformly as a new name "two-hot soft label" (including in Figure 1), and providing comparison with the typical soft label in Appendix D.1 (Reviewers r1aS and dTK9)

3) Simplifying Figure 2 to a non-class-wise format (Reviewer 9r4X)

4) Adding two new SOTAs for comparison in Figure 5 and Table 2 (Reviewer dTK9)

5) Providing discussion and comparison with synthetic data-based methods in Appendix C.3 (Reviewer QPXH)

6) Providing comparison with inference time defense method MI in Appendix C.4 (Reviewer r1aS)

7) Providing discussion about the relatively less improvement of TRADES + DUCAT in Appendix D.2 (Reviewers 9r4X and r1aS)

8) Providing clean accuracy of standard training for reference in Appendix E (Reviewer dTK9)

Please kindly let us know if further clarification is required. We would be glad to hear from you. Thank you all again :)

---

### Public Comment · ~Seungju_Cho1 · 2024-11-29
**Question on your paper.**

I found the performance results in this paper very impressive, which motivated me to take a closer look at both the methodology and the provided code. While the proposed approach of introducing dummy classes and using projection for adversarial sample recovery is innovative, I have some questions regarding the evaluation protocol for robustness, particularly under the white-box attack setting.

From what I understand, the projection mechanism used during inference—where predictions in dummy classes are mapped back to original classes—is described as being independent of the computation graph and implemented in a runtime-only manner. If this is the case, would an attacker generating white-box attacks be fully aware of the projection process? In a strict white-box scenario, the attacker is assumed to have complete knowledge of the model, including all components and mechanisms. However, it seems that the projection mechanism might not be considered when crafting adversarial examples. This could potentially lead to overestimating the method’s robustness, as the attacks may only target the dummy classification without accounting for how predictions are ultimately projected back to the original classes.

For example, if an adversarial attack targets a sample originally classified as class 1 and perturbs it such that it falls into dummy class 11 (the dummy class corresponding to class 1), the projection mechanism would map it back to class 1 during inference. This could make it appear that the model is robust, even though the attack successfully fooled the dummy classifier. If the attacker is unaware of this projection step, it could inflate the perceived robustness.

Could you clarify whether the projection mechanism was explicitly included in the adversarial attack process during evaluation? If not, would the results be more representative of a gray-box attack scenario rather than a full white-box setting? Additionally, if the attacks primarily target the dummy classes without considering the projection step, could this explain the high robustness values reported in the paper?

Thank you for your time and clarification!

---

> ### Author Response · Authors · 2024-11-29
> **Response to Public Question**
>
> Thank you for being interested in our work and for regarding our results to be "very impressive". It's a good question with multiple underlying points to discuss.
>
> Firstly, we would rather interpret the "complete knowledge of the model" in the "strict white-box scenario" as model architecture, parameters, gradients, and training dataset just as the original definition [1, 2, 3] adopted by most previous works. We respectfully disagree that a runtime projection with a customizable specific mechanism should belong to the knowledge, just as we detailed in _**the threat models part of Section 3.1**_. The key point is that the real-world dangerousness of white-box attacks mainly appears when the attacker acquires a _**shadow model**_ of the target model in advance (e.g., by pretraining foundation surrogate models or by model extraction attack on the target model) [4, 5, 6]. However, to the best of our knowledge, through any of such existing methods, it is _**not possible**_ to reproduce our runtime projection in the shadow model. Please kindly remember that, for works relevant to security like ours, instead of indiscriminately referring to a simplified definition only in _**literal meaning**_, we should always keep in mind what is the _**real-world threat it exactly intended to represent**_, in order not to give the attacker a capability it can actually not possess [7].
>
> Secondly, despite the disagreement, let's still simply see what happens even if the attacker is additionally aware of the projection. Please kindly note that it is actually not "the attacks target the dummy classification". Instead, it is our defense that _**actively induces**_ the attacks to perturb benign samples to the one-to-one corresponding dummy classes. So for _**untargeted attacks**_, it would _**change nothing**_ even if the attacker knows the specific projection because the attacks can not escape from our defense trap without giving a specific target apart from the original class and dummy class of the current sample. While for _**targeted attacks**_, they could indeed utilize the projection to compulsively appoint other attack targets to try to bypass our defense. However, the fact is that targeted attacks are simply neither sufficiently dangerous in general nor commonly adopted compared with untargeted ones in real-world practice [8, 9]. This is quite natural because for most samples, compared with those untargeted attacks, targeted attacks abandon the easiest target (whether dummy or not) and turn to a harder one. Actually, in our experiments, targeted attacks knowing the projection are _**even less dangerous**_ than untargeted ones without knowing that. We had considered and experimentally demonstrated this in _**Appendix C.1**_ of our manuscript, please kindly refer to it for more details.
>
> Finally, we also respectfully disagree that the projection "makes it appear that the model is robust, even though the attack successfully fooled the dummy classifier". First, as we explained above, it does not actually "fool" the dummy classifier as it complacently believes. On the contrary, it actually falls into our well-designed defense trap. Second, what's important should not be what happens within the dummy model, but _**whether the whole system or application becomes more robust**_ or not. Otherwise, would you also question the other works beyond the conventional AT paradigm, such as the inference time approach MI [10] suggested by our kind reviewer which does even not change the model itself for robustness? We respectfully do not think that would be fair.
>
> We sincerely hope our clarification would be helpful for your question. We are always here for further discussion if needed. Thank you :)
>
>
> [1] Szegedy et al. Intriguing properties of neural networks. In ICLR, 2014.
>
> [2] Goodfellow et al. Explaining and harnessing adversarial examples. In ICLR, 2015.
>
> [3] Papernot et al. Practical black-box attacks against machine learning. In Asia CCS, 2017.
>
> [4] He et al. Model extraction and adversarial transferability, your BERT is vulnerable! In NAACL, 2021.
>
> [5] Qin et al. Training meta-surrogate model for transferable adversarial attack. In AAAI, 2023.
>
> [6] Zhang et al. Introducing foundation models as surrogate models: Advancing towards more practical adversarial attacks. arXiv preprint arXiv:2307.06608.
>
> [7] Fenaux et al. Analyzing adversarial examples: A framework to study adversary knowledge. arXiv preprint arXiv:2402.14937.
>
> [8] Rathore et al. Untargeted, targeted and universal adversarial attacks and defenses on time series. In IJCNN, 2020.
>
> [9] Cai et al. Ensemble-based black-box attacks on dense prediction. In CVPR, 2023.
>
> [10] Pang et al. Mixup inference: Better exploiting mixup to defend adversarial attacks. In ICLR, 2020.

---

> ### Public Comment · ~Seungju_Cho1 · 2024-11-29
> **Response to authors.**
>
> This work does not seem to fully align with the strict definition of a white-box defense setting. It might be more appropriate to compare the proposed method with defenses designed for similar settings. Since the evaluation does not consider a worst-case adversarial scenario where the attacker has full knowledge, these comparisons could be somewhat unclear.
>
> I would be interested to hear how other reviewers view these points.
>
>
> **Concerns**
>
> This paper does not adhere to the standards outlined in [1] for proper evaluation of defenses. For example, Section 7.2, *Recommendations for Defense*, in [1] suggests conducting adaptive and fully white-box attacks to evaluate the robustness of defense methods. The current paper does not align with these criteria.
>
> Specifically, I believe this method could be easily circumvented if an attacker is aware of the defender's dummy class trick. For instance, when attacking class 1, the attacker could target both class 1 and the dummy class (e.g., class 11) simultaneously, rendering the defense ineffective.
>
> The reason this approach is not more effective than existing methods is straightforward: it assumes the attacker is unaware of the defender’s strategy. However, as shown in the appendix, the performance under targeted attacks is lower than PGD-AT (73.02 robustness by PGD-AT, 70.77 by proposed method. Here, the argument that targeted attacks are less important due to their lower attack success rates may not be entirely convincing. The proposed method performs worse than the most fundamental baseline, PGD-AT. This suggests that the proposed approach could be more vulnerable than the baseline when facing more sophisticated adaptive attacks.
> ), which supports the argument that this method may not be robust when faced with adaptive attacks.
>
> To provide another example, in a binary classification task (e.g., distinguishing between classes 0 and 1) with dummy classes 2 and 3 added, if an attacker attempts to misclassify data from class 0 by ensuring it is not classified as either 0 or 2, it is unclear whether this defense mechanism would remain effective.
>
> Given these observations, this defense method seems more suitable for comparison with black-box or gray-box defense strategies, rather than adversarial training methods designed for full white-box scenarios.
>
> Finally, because this is not a complete white-box defense, I believe it would not meet the rigorous evaluation standards of platforms such as [RobustBench](https://robustbench.github.io/), which specifically assess adversarial training methods.
>
> My concern is that, if this paper is accepted without clarifying its limitations in white-box settings, it may cause confusion among adversarial robusteness researchers. This could lead to misunderstandings regarding the applicability and effectiveness of this method in truly white-box environments. I hope the authors can address these issues to provide greater clarity and avoid potential misinterpretations.
>
> [1] Carlini, Nicholas, and David Wagner. "Adversarial examples are not easily detected: Bypassing ten detection methods." Proceedings of the 10th ACM workshop on artificial intelligence and security. 2017.

---

> ### Author Response · Authors · 2024-11-29
> **Response to Public Comment**
>
> Thank you for your further comments. However, we sincerely believe we have already responded to all these concerns above.
>
> We would also like to conclude our views for the kind reference of our official reviewers.
>
> - Proposing a novel paradigm unseen before, whether or not our work aligns with white-box definition should not depend on the previous _**literal expression**_ which could not have anticipated our work, but depend on _**the exact real-world threat that the definition exactly intended to represent**_. So we respectfully disagree this work is not aligned with white-box definition.
>
> - We are always honest with the case in targeted attacks (the so-called "worst-case adversarial scenario") from the first version of our submission. It is a widely recognized fact that targeted attacks are _**less dangerous and less commonly used**_ than untargeted attacks in real-world practices, so our results on targeted attacks are not as good as on untargeted ones would _**not significantly impact our main contributions**_.
>
> We would be glad to have further discussion if there are still any specific points to be clarified. Thank you.
>
> ---
> ---
>
> **Modified:**
>
> Thank you for further modifying the previous comments by adding the detailed concerns. Accordingly, below we also further detail our response.
>
> Firstly, we fully understand that, for works trying to _**propose novel paradigms unseen before**_ like ours, there might be such concerns about the alignment with previous settings. As detailed in the last response, we sincerely believe that, such concerns in AI security area should be discussed based on _**the exact real-world threat that the settings intended to represent**_. Based on this opinion, we have already discussed in the last response why our work is well-aligned with white-box definition, achieving white-box robustness.
>
> Furthermore, even if you would like to _**discuss the literal meaning of white-box definition**_, we would like to suggest referring to our references [1, 2, 3], all of which limit white-box knowledge within model architecture, parameters, gradients, and training dataset, _**excluding our run-time strategy**_. They are _**more persuasive**_ than yours [1] because our [1, 2] is the first two works of white-box adversarial attacks, and [3] is the first work explicitly distinguishing white-box and black-box knowledge for adversarial attacks. On the contrary, your reference [1] even aims at adversarial example detectors instead of standard classifiers.
>
> Secondly, we would like to respectfully point out that, your other objection points apart from the ones regarding the white-box definition seem also _**not fair enough to us**_. Specifically, we sincerely hope that you could kindly reconsider the following points:
>
> - The suggested circumvention approach not only assumes a real-world unpractical attacker capability but even also tries to modify standard attacks to customize new attacks against our defense, which is obviously _**misaligned with the fair and general evaluation**_ on the AT benchmarks.
>
> - It is _**not just our argument**_ that targeted attacks are _**secondary**_ compared with untargeted ones, it is _**a widely recognized fact**_ in the adversarial area supported by a number of previous works such as [8, 9] above.
>
> - We are sad that you only refer to PGD-AT for performance under targeted attacks, which is _**the worst one**_ among the results on all the four AT benchmarks we provided in _**Appendix C.1**_. We would appreciate it if you could _**fairly consider the four results together**_, through which it could be easily found that even under targeted attacks (the so-called "worst-case adversarial scenario"), our method is at least _**still competitive**_ to the benchmarks.
>
> Finally, we are frustrated that your comment seems to censure our work on the lack of "clarifying its limitations in white-box settings", while we are always honest with it from the first version of our submission. We provided an independent _**Threat models**_ subsection (which is not commonly seen in AT works) in _**Section 3.1 Experimental Setup**_ to clarify the white-box settings and explain its reasonability. Also, we had anticipated the potential objection like yours and consequently provided competitive experimental results (again, please kindly refer to all benchmarks instead of only PGD-AT) that _**specifically turn to the scenario with targeted attacks**_ (again, the "worst-case"). Still, we are sincerely willing to refine our manuscript if you have any specific suggestions on how we could do better in the presentation of this clarification.
>
> We would always be open to further discussions if needed. Thank you.

---

### Meta-Review · Area_Chair_HtK5 · 2024-12-26

**Metareview:**

This paper aims to reduce the tradeoff between clean and robust accuracy under adversarial attacks. To achieve this, it introduces an adversarial training method with two-hot soft labels, where a dummy set of classes is added to the classification task to relax the constraint that adversarial and clean samples be classified to the same class. The method is evaluated on a range of datasets and baselines and is shown to significantly improve clean and robust accuracy.

Strengths:
- Novel two-hot-soft-label approach
- The method presented can be integrated into other adversarial training methods and is motivated by an interesting empirical analysis.
- Extensive evaluation against different baselines on several datasets and different network architectures with sometimes significant improvements
- Paper is well written

Weaknesses:
- Lack of evaluation with an adaptive attack with full knowledge of the defense
- Lack of comprehensive evaluation with state-of-the-art methods that incorporate synthetic data

Overall, while the paper presents a novel approach with promising results, there is lack of evaluation with an adaptive attack with full knowledge of the defense (in the paper the runtime projection is not considered), which may result in an overestimate of the robustness gains (as in Tramèr et al 2000); the results shown in the targeted attack setting (Appendix C.1) already hint at this possibility. Given the significant improvements claimed, the AC believes it is necessary for the method to perform well in such an evaluation for the paper to be accepted. The authors are recommended to include this evaluation and resubmit to a future venue.

**Additional Comments On Reviewer Discussion:**

There was extensive discussion on several points raised by reviewers
- Clarification of the novelty with regards to other methods using soft labels
- Marginal improvements compared when incorporated with TRADES
- Comparison to other SOTA methods
- Overall framing of the paper

The authors clearly differentiated their method from other soft-label approaches to the satisfaction of the reviewers, and included a discussion on performance with TRADES. They also included some additional comparisons with SOTA methods, though a full analysis was not included for methods using synthetic data due to limited time. Finally, the claims in the paper were slightly toned down in response to reviewer feedback. Multiple reviewers raised their scores in response to the rebuttal, but the paper still remained borderline.

More significantly, there was a public comment highlighting that the attacks evaluated did not incorporate knowledge of the runtime projection layer. While the authors argue that they have explicitly mentioned this in their threat model and that it respects the white-box setting, the AC does not quite agree with this characterization. To the authors' credit they are explicit about this, but as previous work has argued (Tramèr et al 2000), it is important for defenses to be evaluated against specifically designed attacks. As mentioned above, it would be prudent to evaluate the proposed defense against such an attack to demonstrate its robustness and validate the significant gains shown against standardized evaluations like AutoAttack.

---

### Decision · Program_Chairs · 2025-01-22

Reject